# The *cis*-regulatory effects of modern human-specific variants

Carly V Weiss[1†], Lana Harshman[2,3†], Fumitaka Inoue[2,3‡], Hunter B Fraser[1], Dmitri A Petrov[1]*, Nadav Ahituv[2,3]*, David Gokhman[1]*

[1]Department of Biology, Stanford University, Stanford, Stanford, United States; [2]Department of Bioengineering and Therapeutic Sciences, University of California San Francisco, San Francisco, San Francisco, United States; [3]Institute for Human Genetics, University of California San Francisco, San Francisco, San Francisco, United States

*For correspondence:
dpetrov@stanford.edu (DAP);
nadav.ahituv@ucsf.edu (NA);
dgokhman@stanford.edu (DG)

[†]These authors contributed equally to this work

Present address: [‡]Institute for the Advanced Study of Human Biology (WPI-ASHBi), Kyoto University, Kyoto, Japan

Competing interests: The authors declare that no competing interests exist.

**Abstract** The Neanderthal and Denisovan genomes enabled the discovery of sequences that differ between modern and archaic humans, the majority of which are noncoding. However, our understanding of the regulatory consequences of these differences remains limited, in part due to the decay of regulatory marks in ancient samples. Here, we used a massively parallel reporter assay in embryonic stem cells, neural progenitor cells, and bone osteoblasts to investigate the regulatory effects of the 14,042 single-nucleotide modern human-specific variants. Overall, 1791 (13%) of sequences containing these variants showed active regulatory activity, and 407 (23%) of these drove differential expression between human groups. Differentially active sequences were associated with divergent transcription factor binding motifs, and with genes enriched for vocal tract and brain anatomy and function. This work provides insight into the regulatory function of variants that emerged along the modern human lineage and the recent evolution of human gene expression.

## Introduction

The fossil record allows us to directly compare skeletons between modern humans and their closest extinct relatives, the Neanderthal and the Denisovan. From this we can make inferences not only about skeletal differences, but also about other systems, such as the brain. These approaches have uncovered a myriad of traits that distinguish modern from archaic humans. For example, our face is flat with smaller jaws, our development is slower, our pelvises are narrower, our limbs tend to be slenderer, and our brain differs in its substructure proportions (*Neubauer et al., 2018*; *Gunz et al., 2019*; *Aiello and Dean, 2002*) (especially the cerebellum; *Kochiyama et al., 2018*). Despite our considerable base of knowledge of how modern humans differ from archaic humans at the phenotypic level, we know very little about the genetic changes that have given rise to these phenotypic differences.

The Neanderthal and the Denisovan genomes provide a unique insight into the genetic underpinnings of recent human phenotypic evolution. The vast majority of genetic changes that separate modern and archaic humans are found outside protein-coding regions, and some of these likely affect gene expression (*Yan and McCoy, 2020*). Such regulatory changes may have a sizeable impact on human evolution, as alterations in gene regulation are thought to underlie most of the phenotypic differences between closely related groups (*Britten and Davidson, 1971*; *King and Wilson, 1975*; *Enard et al., 2014*; *Fraser, 2013*). Indeed, there is mounting evidence that many of the noncoding variants that emerged in modern humans have altered gene expression in cis, shaped phenotypes, and have been under selection (*Yan and McCoy, 2020*; *McCoy et al., 2017*; *Petr et al., 2019*; *Gokhman et al., 2020*; *Colbran, 2019*; *Gokhman et al., 2019*; *Dannemann and*

*Racimo, 2018*; *Weyer and Pääbo, 2016*; *Vespasiani et al., 2020*; *Grogan and Perry, 2020*). Fixed variants, in particular, could potentially underlie phenotypes specific to modern humans, and some of these variants might have been driven to fixation by positive selection.

Unfortunately, our ability to infer the regulatory function of noncoding variants is currently limited (*Chatterjee and Ahituv, 2017*). In archaic humans, incomplete information on gene regulation is further exacerbated by the lack of RNA molecules and epigenetic marks in these degraded samples (*Yan and McCoy, 2020*). We have previously used patterns of cytosine degradation in ancient samples to reconstruct whole-genome archaic DNA methylation maps (*Gokhman et al., 2020*; *Gokhman et al., 2014*; *Gokhman et al., 2016*). However, despite various approaches to extract regulatory information from ancient genomes (*Yan and McCoy, 2020*; *Colbran, 2019*; *Gokhman et al., 2016*; *Barker et al., 2020*; *Batyrev et al., 2019*; *Pedersen et al., 2014*; *Silvert et al., 2019*; *Moriano and Boeckx, 2020*), our understanding of gene regulation in archaic humans remains minimal, with most archaic regulatory information being currently inaccessible (*Yan and McCoy, 2020*). Additionally, whereas expression quantitative trait locus (eQTL) mapping can be used to identify variants that drive differential expression between individuals, it can only be applied to loci that are variable within the present-day human population. Therefore, fixed noncoding variants are of particular interest in the study of human evolution, but are also particularly difficult to characterize.

Massively parallel reporter assays (MPRAs) provide the ability to interrogate the regulatory effects of thousands of variants *en masse* (*Inoue and Ahituv, 2015*). By cloning a candidate regulatory sequence downstream to a short transcribable sequence-based barcode, thousands of sequences and variants can be tested for regulatory activity in parallel. Thus, MPRA is an effective high-throughput tool to identify variants underlying divergent regulation, especially in organisms where experimental options are limited (*Tewhey et al., 2016*; *Klein et al., 2018*; *Ryu et al., 2018*; *Uebbing et al., 2021*). Here, we conducted a lentivirus-based MPRA (lentiMPRA; *Gordon et al., 2020*) on the 14,042 fixed or nearly fixed single-nucleotide variants that emerged along the modern human lineage. We generated a library of both the derived (modern human) and ancestral (archaic human and ape) sequences of each locus and expressed them in three human cell types: embryonic stem cells (ESCs), neural progenitor cells (NPCs), and primary fetal osteoblasts. By comparing the transcriptional activities of each pair of sequences, we generated a comprehensive catalog providing a map of sequences capable of promoting expression and those that alter gene expression. We found that 1791 (13%) of the sequence pairs promote expression and that 407 (23%) of these active sequences drive differential expression between the modern and archaic alleles. These differentially active sequences are associated with differential transcription factor (TF) binding affinity and are enriched for genes that affect the vocal tract and brain. This work provides a genome-wide catalog of the *cis*-regulatory effects of genetic variants unique to modern humans, allowing us to systematically interrogate recent human gene regulatory evolution.

## Results

### LentiMPRA design and validation

To define a set of variants that likely emerged and reached fixation or near fixation along the modern human lineage, we took all the single-nucleotide variants where modern humans differ from archaic humans and great apes (based on three Neanderthal genomes [*Prüfer et al., 2014*; *Prüfer et al., 2017*; *Mafessoni et al., 2020*], one Denisovan genome [*Meyer et al., 2012*], and 114 chimpanzee, bonobo, and gorilla genomes [*de Manuel et al., 2016*]). We excluded any polymorphic sites within modern humans (in either the 1000 Genomes Project [*Auton et al., 2015*] or in dbSNP [*Sherry et al., 2001*]), or within archaic humans and great apes (*Prüfer et al., 2014*; *Prüfer et al., 2017*; *Mafessoni et al., 2020*; *Meyer et al., 2012*; *de Manuel et al., 2016*) (see 'Materials and methods'). The resulting set of 14,042 variants comprises those changes that likely emerged and reached fixation or near fixation along the modern human lineage (*Supplementary file 1a-c*). The vast majority of these variants are intergenic (*Figure 1—figure supplement 1a*). By definition, this list does not include variants that introgressed from archaic humans into modern humans and spread to detectable frequencies. We refer to the derived version of each sequence as the *modern human sequence* and the ancestral version as the *archaic human sequence*.

We analyzed variants that likely emerged and reached fixation or near fixation along the modern human lineage (yellow) and that were not polymorphic in any other ape or archaic genome (green) (top). The modern and archaic human variants and their surrounding 200 bp were synthesized, cloned into barcoded expression constructs, and infected in triplicates into three human cell lines using a chromosomally integrating vector, following the lentiMPRA protocol (*Gordon et al., 2020*) (see 'Materials and methods'). We compared the activity (RNA/DNA) of the modern and archaic human constructs to identify variants promoting differential expression using MPRAnalyze (*Ashuach et al., 2019*) (bottom).

We synthesized a library composed of 200 bp sequences (due to oligonucleotide synthesis length limitations) per each of the 14,042 variants (one sequence for the modern human allele and one for the archaic human allele, *Figure 1—figure supplement 1a–c*). Each sequence contained at its center either the modern or archaic human variant. Out of 14,042 sequence pairs, 13,680 (90%) had a single variant separating the human groups. For the 1362 sequence pairs containing additional variants within the 200 bp window, we used either the modern-only or archaic-only variants throughout the sequence. We amplified this library of sequences, each along with a minimal promoter (mP) and barcode. We then inserted these constructs into the lentiMPRA vector, so that the barcode, which is the readout of activity, is located within the 5'UTR of the reporter gene and is transcribed if the assayed sequence is an active regulatory element (*Gordon et al., 2020*). We associated each sequence with multiple barcodes to achieve a high number of independent replicates of expression per sequence, thereby reducing potential site-of-integration effects. 97% of sequences had at least 10 barcodes associated with them, with a median of 96 barcodes per sequence (*Figure 1—figure supplement 2a*). Furthermore, we used a chromosomally integrating construct rather than an episomal construct due to the improved technical reproducibility and correlation of results from chromosomally integrating constructs with functional genomic signals like TF ChIP-seq and histone acetylation marks (*Inoue et al., 2017*). To further reduce lentivirus site-of-integration effects, this vector contained antirepressors on either side and was integrated in multiple independent sites, with each sequence marked by multiple barcodes (see Discussion for additional lentiMPRA limitations). Importantly, despite the caveat of interrogating sequences outside of their endogenous context, MPRAs were shown to generally replicate the endogenous activity of sequences (*Inoue et al., 2017*; *Klein et al., 2020*; *Kircher et al., 2019*).

The brain and skeleton have been the focus of evolutionary studies due to their extensive phenotypic divergence among human lineages (*Aiello and Dean, 2002*). Therefore, we chose human cells related to each of these central systems: NPCs and primary fetal osteoblasts. In addition, we used ESCs (line H1, from which the NPCs were derived) to gain insight into early stages of development. Finally, the abundance of previously published regulatory maps for these three cell types (*Gokhman et al., 2014*; *Ernst and Kellis, 2012*; *Kundaje et al., 2015*) also enables the investigation of the dynamics of evolutionary divergence at different regulatory levels. While these cell types represent diverse systems, further studies are needed in order to characterize the activity of these sequences in other cell types.

We used the library of 14,042 pairs of archaic and modern human sequences, together with positive and negative control sequences, to infect each cell type. As positive controls for ESCs and NPCs, we added a set of 199 sequences with known regulatory capacity from previous MPRAs (*Supplementary file 1d*). To our knowledge, there have not been any MPRAs conducted in osteoblasts, so we searched the literature for putative regulatory regions in osteoblasts and other bone cell types and used these as putative positive controls (*Supplementary file 1d*, see 'Materials and methods'). As negative controls, in all cell types, we randomly chose 100 sequences from the library and scrambled the order of their bases, creating a set of GC content matching sequences that had not been previously established to drive expression (*Supplementary file 1e*).

We performed three replicates of library infection in each cell type and quantified barcode abundance for each sequence in RNA and DNA (*Figure 1*). To assess the reproducibility of our lentiMPRA results, we calculated the RNA/DNA ratio (a measure of expression normalized to the number of integrated DNA molecules) for each sequence and compared it across the three replicates per cell type. We saw a strong correlation of RNA/DNA ratios between replicates for all cell types (Pearson's $r = 0.76$–$0.96$, $p<10^{-100}$, *Figure 1—figure supplement 2b*), with the lower correlation scores being in ESC, likely due to our use of lower multiplicity of infection (MOI) in these cells due to their increased sensitivity to lentivirus infection. High barcode and read coverage in MPRA generally

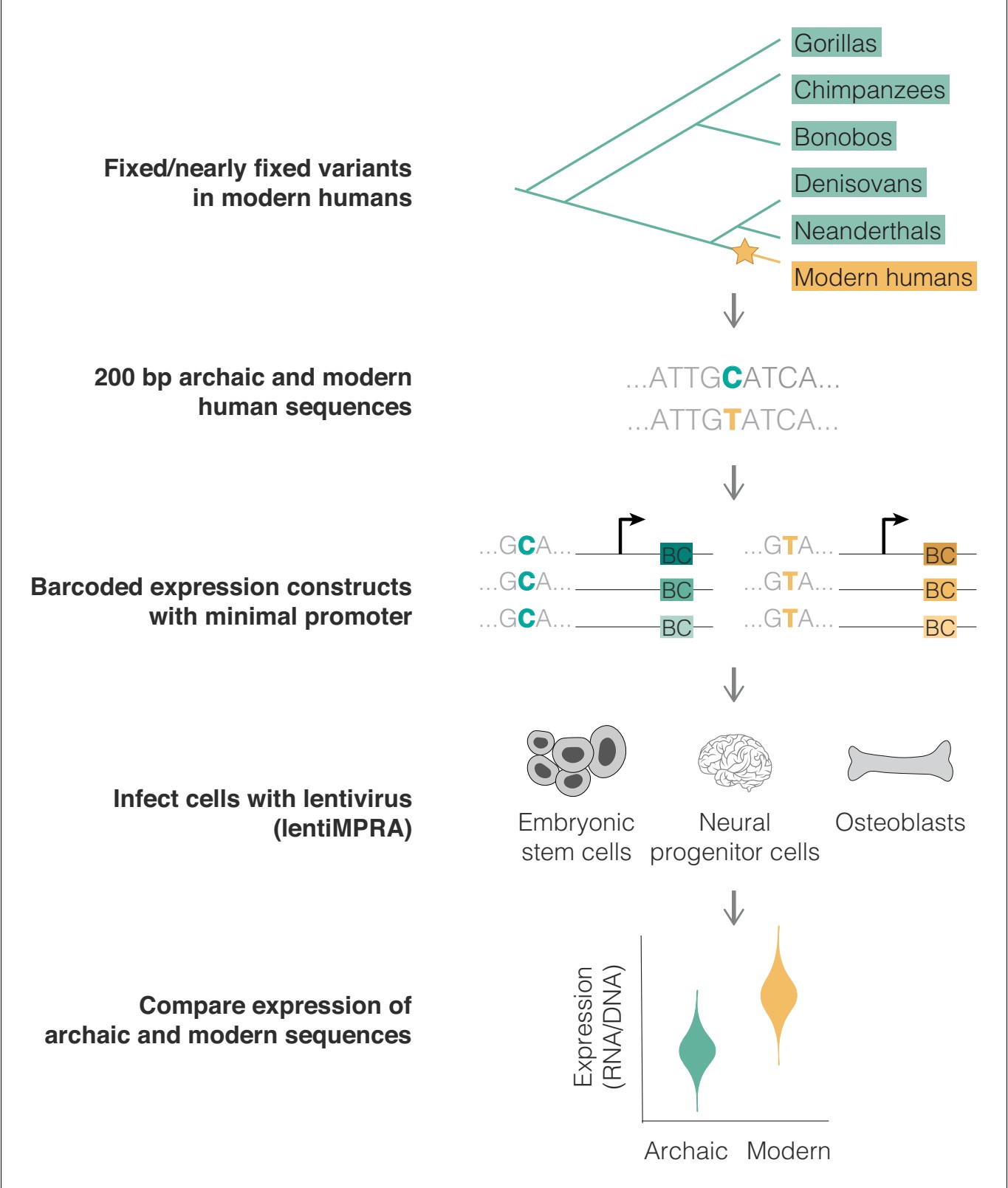

**Figure 1.** Using lentivirus-based MPRA (lentiMPRA) to identify variants driving differential expression in modern humans. The online version of this article includes the following figure supplement(s) for figure 1:

**Figure supplement 1.** Classification of chromHMM annotations for different groups of variants.

*Figure 1 continued on next page*

*Figure 1 continued*

**Figure supplement 2.** Reproducibility of lentivirus-based MPRA (lentiMPRA) data.

provides increased power to detect differences in allelic expression (*Gordon et al., 2020*; *Kircher et al., 2019*). Thus, to determine how variability depended on our barcode counts, we downsampled the number of barcodes per sequence and calculated the RNA/DNA ratio at each step for each of the three replicates. In agreement with previous studies (*Inoue et al., 2017*), we found that the number of barcodes used in this study is well within the plateau, suggesting that the number of barcodes is not a limiting factor in our experiment (*Figure 1—figure supplement 2c*). Finally, we assessed the distribution of RNA/DNA ratios across our scrambled sequences and positive controls. The mean RNA/DNA ratio of the scrambled sequences was lower than that of the positive control sequences in ESCs and NPCs (p=$2.7\times10^{-8}$ for ESCs and p=$1.8\times10^{-6}$ for NPCs, *t*-test, see 'Materials and methods', *Figure 1—figure supplement 2d*), but not in osteoblasts (p=0.25). This is unlikely due to a problem with the osteoblasts, as the osteoblast-related controls show similar expression in all three cell types. Moreover, ESC and NPC positive controls are active in osteoblasts (p=$1.1\times10^{-3}$). The correlation between replicates was also similar between osteoblasts and the other two cell types (*Figure 1—figure supplement 2b*). Thus, the lack of activity of the osteoblast putative positive controls is likely because, in contrast to the ESC and NPC confirmed positive controls, the osteoblast putative positive controls were not previously tested in an MPRA, and some of these putative enhancers were identified in mouse and were not validated in human. Overall, these results suggest that the lentiMPRA was technically reproducible and adequately powered to detect expression.

## Characterization of active regulatory sequences

We first examined which of the assayed sequences are able to drive expression. To do so, we utilized MPRAnalyze (*Ashuach et al., 2019*), which uses a model for each of the RNA and DNA counts, estimates transcription rate, and then identifies sequences driving significant expression. We also added an additional stringency filter whereby a sequence is only considered expressed if it had an RNA/DNA ratio significantly higher than that of the scrambled sequences (false discovery rate [FDR] <0.05). We found that in ESCs, 8% (1183) of sequence pairs drove expression in at least one of the alleles, 6% (814) in osteoblasts, and 4% (602) in NPCs (FDR <0.05, *Supplementary file 1a-c*, *Figure 1—figure supplement 2d*, see 'Materials and methods'). Hereinafter, we refer to these sequences as *active* sequences. Overall, 13% (1791) of archaic and modern human sequence pairs were active in at least one cell type, 4% (586) in at least two cell types, and 2% (222) in all three cell types (overlap of 75-fold higher than expected, p<$10^{-100}$, Super Exact test; *Wang et al., 2015*, *Figure 2a*).

Some of these sequences may show activity in the lentiMPRA experiment but not in their endogenous genomic context. To test whether activity in our lentiMPRA reflects true biological function, we investigated whether our active sequences had expected regulatory characteristics in the modern human genome. Active regulatory sequences in the genome tend to bear active chromatin marks. Therefore, we examined whether active sequences in lentiMPRA tend to be enriched for markers of active chromatin in their endogenous context. We first tested overlap with five histone modification marks and one histone variant associated with active chromatin (H3K27ac, H3K4me1, H3K4me2, H3K4me3, H3K9ac, and H2A.Z), as well as with two histone modification marks associated with repressed chromatin (H3K9me3 and H3K27me3, see 'Materials and methods') (*Kundaje et al., 2015*). We found that on average, active sequences were 1.6- to 2.7-fold more likely than inactive sequences to have active chromatin marks, depending on cell type. Also, these sequences tended to show relatively fewer repressive marks compared to active marks (*Figure 2b–d*, *Supplementary file 2*). These trends get stronger when looking at more highly active sequences. For example, while only 18% of inactive sequences in ESCs overlap H3K4me2 peaks, 70% of active sequences with an RNA/DNA ratio ≥3 in ESCs overlap H3K4me2 peaks (FDR = $4.4\times10^{-16}$, Fisher's exact test, *Figure 2b–d*, *Supplementary file 2*). To further test the functional characteristics of active sequences, we analyzed chromHMM annotation (*Ernst and Kellis, 2012*; *Kundaje et al., 2015*), which uses chromatin signatures to subdivide the genome into functional regions. Of the 14,042

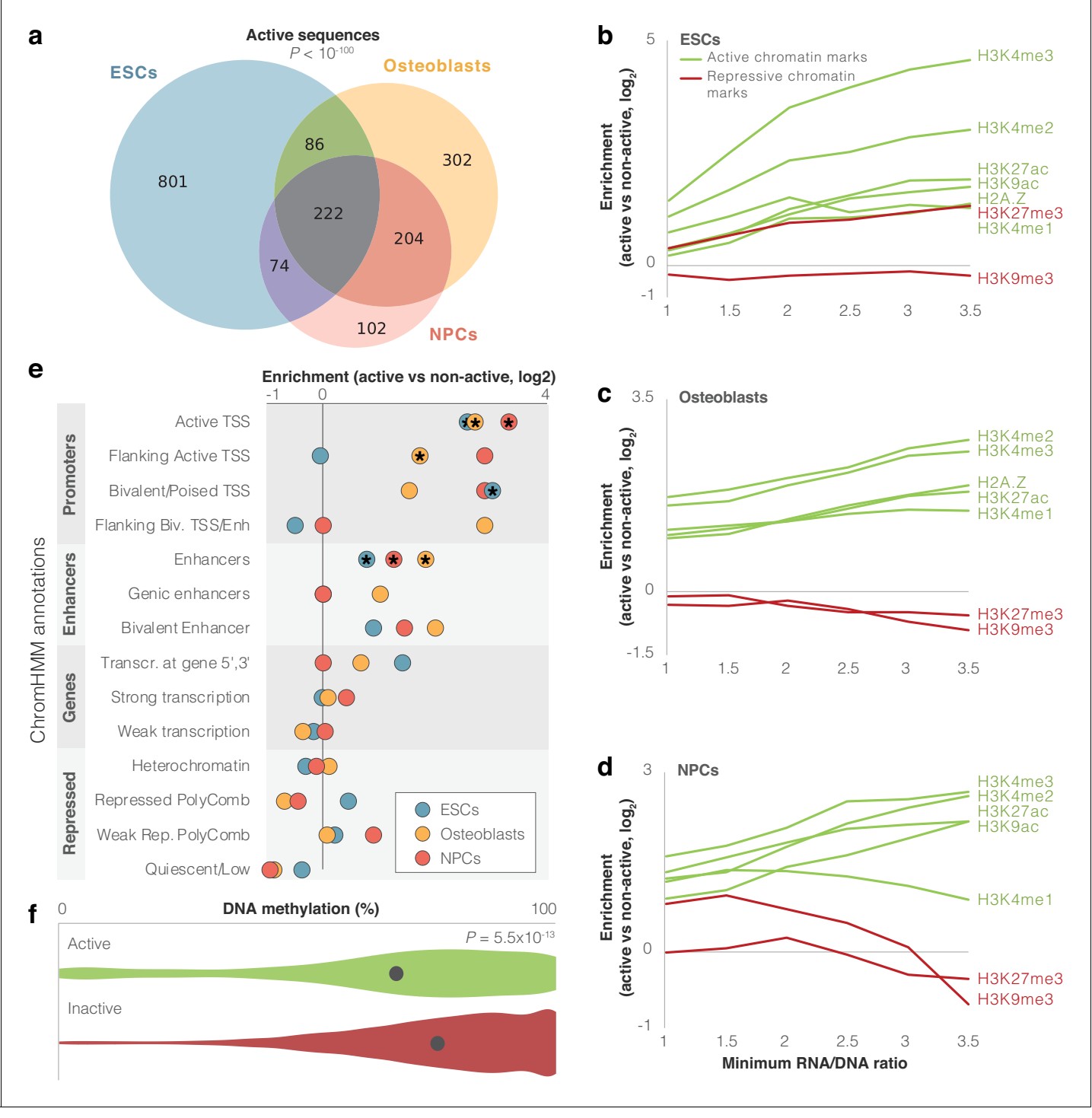

**Figure 2.** Identification of modern human sequences promoting expression in lentivirus-based MPRA (lentiMPRA). (**a**) Overlap between cell types of active sequences. Super Exact test p-value is shown for the overlap of the three groups. (**b-d**) Enrichment levels of active and repressive histone modification marks within active sequences. Enrichment is computed compared to inactive sequences. The enrichment of H3K27me3 in embryonic stem cells (ESCs) possibly reflects the presence of this mark in bivalent genes, which become active in later stages of development (*Blanco et al., 2020*). For confidence intervals, see *Supplementary file 2*. (**e**) Enrichment of differentially active sequences in various chromatin-based genomic annotations. Missing circles reflect no differentially active sequences in that category. Stars mark significant enrichments (false discovery rate [FDR] <0.05). (**f**) Violin plots of DNA methylation levels for active (green) vs. inactive (red) sequences in osteoblasts. Methylation levels per sequence were computed as the mean methylation across all modern and archaic human bone methylation samples. The circle marks mean methylation across all sequences in each group. *t*-test p-value is shown.

*Figure 2 continued on next page*

*Figure 2 continued*

The online version of this article includes the following figure supplement(s) for figure 2:

**Figure supplement 1.** Differential expression is replicated across overlapping sequences and in a reporter assay validation.

sequences, 2,163 (15%) overlapped promoter or enhancer chromHMM annotations in at least one of the three cell types. Additional 2658 sequences (19%) overlapped such marks in other cell types not included in this study. Compared to inactive sequences, we found that active sequences are enriched for promoter and enhancer marks (FDR <0.05 in each of the cell types for overlap with *active TSS* and *enhancers*, *Figure 2e*, *Figure 1—figure supplement 1*, *Supplementary file 1f*, *Supplementary file 2*). We also found that compared to inactive sequences, active sequences are 6–32% closer to GTEx (*GTEx Consortium, 2015*) eQTLs, depending on cell type (FDR <0.05, *t*-test). Active sequences are also 1.2–1.3× closer to transcription start sites (TSS), with 32–39% of them located within 10 kb of a TSS, depending on cell type (FDR <0.05, *t*-test, *Supplementary file 2*).

Active genomic regions often show reduced DNA methylation levels compared to inactive regions (*Jones, 2012*). To further test if the activity we detected in the lentiMPRA reflects true biological function, we tested whether the active sequences in the lentiMPRA tend to be hypomethylated in their endogenous genomic context. To do so, we used our previously published modern and archaic human DNA methylation maps (*Gokhman et al., 2020*; *Gokhman et al., 2014*; *Gokhman et al., 2016*). Because the DNA methylation maps originate from skeletal samples, we compared them to the osteoblast lentiMPRA data. We found that active sequences are significantly hypomethylated compared to inactive sequences (p=$5.5\times10^{-13}$, *t*-test, *Figure 2f*) and that their activity level (RNA/DNA ratio) is negatively correlated with methylation levels ($6.0\times10^{-9}$, Pearson's $r = -0.24$).

Finally, compared to inactive sequences, active sequences show slightly higher sequence conservation in primates, indicating a potential functional role (PhyloP, −0.05 on average for inactive, −0.04 for active, FDR = $1.1\times10^{-3}$, *t*-test) with more highly active sequences showing higher conservation levels (e.g., 0.24 for active sequences with RNA/DNA ratio ≥4, *Figure 2—figure supplement 1a*, *Supplementary file 2*). In summary, we found that sequences that are capable of driving expression tend to overlap active chromatin marks, are depleted of repressive chromatin marks, closer to TSS and eQTLs, and have higher sequence conservation, giving us confidence that the MPRA provides us with biologically meaningful results.

## Differentially active sequences between modern and archaic humans

We next set out to identify modern and archaic human sequences driving differential expression. We used MPRAnalyze (*Ashuach et al., 2019*) to compare expression driven by the modern and archaic sequences. Out of the active sequence pairs in each cell type, 110 (9%) in ESCs drive significantly differential expression between modern and archaic humans, 243 (30%) in osteoblasts, and 153 (25%) in NPCs (FDR ≤ 0.05, see Materials and methods, *Figure 3a–c*, *Figure 1—figure supplement 2*, see Discussion for cell-type differences). We refer to these sequence pairs hereinafter as *differentially active* sequences. Overall, we see significant overlap between cell types in differentially active sequences: 407 sequences (23% of active sequences) were differentially active in at least one cell type, 89 (5%) in at least two cell types, and 10 (0.6%) in all three cell types (eightfold higher than expected compared to active sequences, p=$5\times10^{-7}$, Super Exact test (*Wang et al., 2015*), *Figure 3d*).

As expected from such closely related organisms, and similar to other MPRAs that compared nucleotide variants (see Discussion), including one that compared human and chimp sequences (*Ryu et al., 2018*), most sequences drove modest magnitudes of expression difference; of the 407 differentially active sequences, the median fold-change was 1.2×, and only five sequences had a fold-change greater than 2× (*Figure 3a–c*). We refer to differentially active sequences where modern human expression is higher/lower than archaic human expression as up/downregulating sequences, respectively. In ESCs and NPCs, sequences were equally likely to be up- or downregulating (51% and 52% of differentially active sequences were downregulating, p=0.92 and 0.63, respectively, binomial test), while in osteoblasts downregulation was observed slightly more often (59%, p=$6.9\times10^{-3}$). Finally, we examined the 89 sequence pairs that were differentially active in two cell

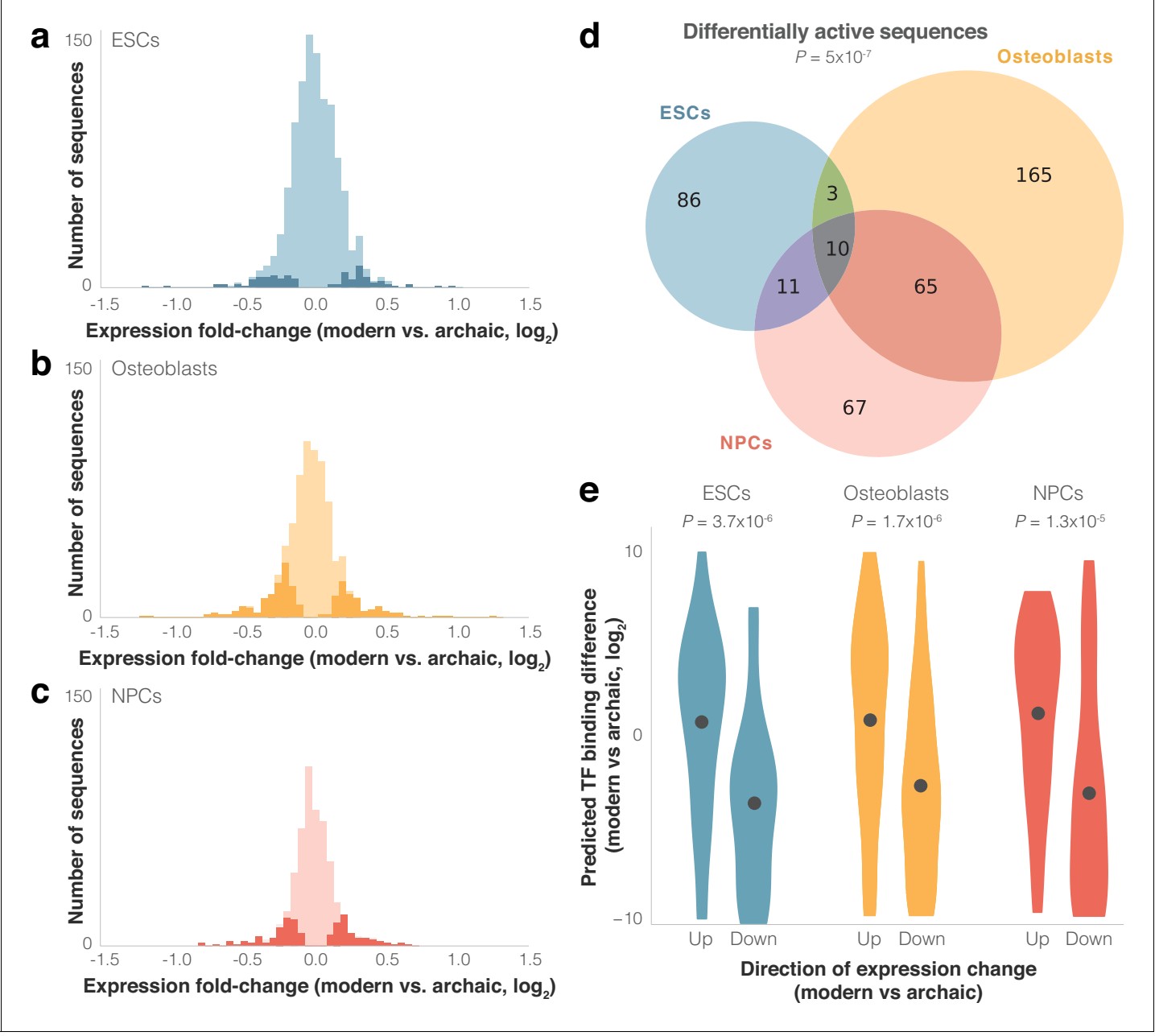

**Figure 3.** Differential activity of derived modern human sequences. (a–c) Distributions of expression fold-changes (RNA/DNA) of active (light) and differentially active (dark) sequences in each cell type. (d) Overlap of differentially active sequences between cell types. Super Exact test p-value is presented for the overlap of the three groups compared to active sequences. In the 10 sequences that were differentially active across all three cells types, the direction of fold-change was identical across all cell types (p=$1.9 \times 10^{-3}$, binomial test). (e) Violin plots of predicted transcription factor (TF) binding score difference between modern and archaic sequences. Positive scores represent increased binding in the modern sequence. Points show mean.

The online version of this article includes the following figure supplement(s) for figure 3:

**Figure supplement 1.** Differential activity is associated with differential DNA methylation and transcription factor (TF) binding.

**Figure supplement 2.** Predicted transcription factor (TF) binding is correlated with differential activity.

types and the 10 sequence pairs that were differentially active in all three cell types, and tested how often the direction of differential activity in one cell type matched the direction in the other cell types. We found a strong agreement in the direction of differential activity across cell types (87 out of 89 of sequence pairs that are differentially active in two cell types, p=$6.5 \times 10^{-24}$, and 10 out of 10

for three cell types, p=9.5×10$^{-7}$, binomial test). We also observed a high correlation between the magnitudes of differential activity (Pearson's $r$ = 0.82, p=1.6×10$^{-27}$). That differentially active sequences from one cell type are predictive of differential activity in other cell types, even of cell types as disparate as those used here, suggests that these sequences are likely to be differentially active in other cell types not assayed in this lentiMPRA.

To further test the replicability of these results, we examined the relationship between pairs of overlapping differentially active sequences (i.e., variants that are <200 bp apart and thus appear in more than one sequence, three overlapping pairs in ESCs, five in osteoblasts, and two in NPCs). We found that the direction of expression change is identical in all pairs of overlapping sequences (p=2.0×10$^{-3}$, binomial test), and that the magnitude of their expression change is highly correlated (Pearson's $r$ = 0.95, 2.4 × 10$^{-5}$, *Figure 2—figure supplement 1b*). To validate these results with an orthogonal method, we tested four differentially active sequences from each cell type in a luciferase reporter assay and found that the direction and magnitude of differential expression tended to replicate the lentiMPRA results (9 out of 12 sequences, Pearson's $r$ = 0.67, p=3.7×10$^{-4}$, *Figure 2—figure supplement 1c*, *Supplementary file 1g*). These results suggest that the lentiMPRA was both technically reproducible across cell types and assays and also indicative of true biological signal.

Finally, we examined the endogenous genomic locations of differentially active sequences, focusing on promoters and enhancers. Between 33% and 45% of these sequences are within 10 kb of a TSS (depending on cell type, *Supplementary file 1h*). Analyzing chromHMM (*Ernst and Kellis, 2012*; *Kundaje et al., 2015*), we found that between 20% and 25% of the differentially active sequences are within putative promoter or enhancer regions (*Supplementary file 1f*). To test if differentially active sequences are enriched within regulatory elements, we compared the proportion overlapping chromHMM promoters and enhancers in differentially active sequences to that proportion in the other active sequences. We found that differentially active sequences are over-represented within putative enhancer regions in NPCs (2.2-fold, FDR = 0.03, Fisher's exact test, *Figure 1—figure supplement 1c,d*). These results support a model of rapid enhancer evolution in modern humans, as previously reported for other mammals (*Villar et al., 2015*) (see 'Discussion').

## Molecular mechanisms underlying differential activity

Next, we sought to understand what regulatory mechanisms might be associated with differential activity. Changes in expression are often linked to changes in regulatory marks. For example, increased DNA methylation tends to be associated with reduced activity (*Jones, 2012*). We therefore tested methylation levels in each pair of sequences and examined if the human group with the lower sequence activity tends to show higher methylation levels. Here too, because the DNA methylation maps originate from bone samples (*Gokhman et al., 2020*; *Gokhman et al., 2014*; *Gokhman et al., 2016*), we compared them to the osteoblast lentiMPRA data. We found that upregulating sequences indeed have a slight but significant tendency to be hypomethylated in modern compared to archaic humans, and that downregulating sequences tend to be hypermethylated in modern compared to archaic humans (on average −2% methylation in upregulating sequences and +1% methylation in downregulating sequences in the modern compared to the archaic genomes, p=0.028, paired *t*-test; *Figure 3—figure supplement 1a*). This trend is slightly more pronounced when looking at the most differentially regulating sequences. For example, the top 10 most downregulating sequences show on average +8% methylation in modern compared to archaic humans, whereas the top 10 most upregulating sequences show −7% methylation in modern compared to archaic humans. We also examined promoter regions (5 kb upstream to 1 kb downstream of a TSS), where the association between methylation and reduced activity is known to be stronger compared to the rest of the genome (*Jones, 2012*). Indeed, we found that upregulating promoter sequences have +5% methylation on average in the modern compared to the archaic genomes, while downregulating promoter sequences have −8% methylation (p=0.034, paired *t*-test; *Figure 3—figure supplement 1b*). This trend is more pronounced in CpG-poor promoters, where the link between methylation and expression is known to be stronger (*Lister et al., 2009*; *Stadler et al., 2011*; *Schlesinger et al., 2013*) (−15% methylation in upregulating sequences and +15% methylation in downregulating promoter sequences in modern compared to archaic humans; p=6×10$^{-3}$, paired *t*-test; *Figure 3—figure supplement 1c*).

We conjectured that some of the differential activity in these loci might have been driven by alterations in TF binding. To investigate this, we compared predicted TF binding affinity to the modern

and archaic sequences using FIMO (*Grant et al., 2011*). We found that (1) compared to other active sequences, the difference in predicted binding between the modern and archaic human alleles tends to be larger for differentially active sequences (combined across cell types: 4.3×, p=0.02, *t*-test, *Figure 3—figure supplement 1d*); (2) the directionality of differential expression tends to match the directionality of differential binding, that is, upregulating sequences tend to have stronger predicted binding for the modern human sequence, whereas downregulating sequences tend to have stronger predicted binding for the archaic sequence (p=$3.7 \times 10^{-6}$ for ESCs, p=$1.7 \times 10^{-6}$ for osteoblasts, and p=$1.3 \times 10^{-5}$ for NPCs, binomial test, *Figure 3e*, see Materials and methods); and (3) the magnitude of expression difference is correlated with the magnitude of predicted binding difference (Pearson's $r = 0.43$ and p=$1.2 \times 10^{-3}$ for ESCs, Pearson's $r = 0.23$ and p=0.02 for osteoblasts, and Pearson's $r = 0.35$ and p=$2.4 \times 10^{-3}$ for NPCs, *Figure 3—figure supplement 2a–c* and *Supplementary file 3*). These results support the notion that alterations in TF binding played a role in shaping some of the expression differences between modern and archaic humans.

To identify the TFs that primarily drove these observations, we investigated which motif changes are most predictive of expression changes. For each TF and the sequences it is predicted to differentially bind, we examined the correlation between binding and expression fold-change (either positive or negative). We found that changes to the motifs of 14 TFs were predictive of expression changes (*Figure 3—figure supplement 2d*, *Supplementary file 3b*). All of these TFs had a positive correlation between changes in their predicted binding affinity and changes in expression of their bound sequences, reflective of their known capability to promote transcription (*Suske, 2017*; *Frey-Jakobs et al., 2018*; *Bruderer et al., 2013*; *Frietze et al., 2010*; *Song et al., 2003*; *Zhu et al., 2018*; *Ji et al., 2020*; *Morita et al., 2016*; *Syafruddin et al., 2020*). Of note, the use of an mP with basal activity in the MPRA design means that transcriptional repression is less likely to be detected, and therefore, further investigation is required in order to identify potential repressive activity in these sequences (see Discussion).

Next, we sought to explore if some motif changes are particularly over-represented within differentially active sequences, suggestive of a more central role in shaping modern human regulatory evolution. To control for sequence composition biases, we used active sequences as a background to search for motif enrichment within differentially active sequences. We found that ZNF281, an inhibitor of neuronal differentiation (*Pieraccioli et al., 2018*), is significantly enriched: out of 153 differentially active sequences in NPCs, 14 are predicted to be bound by ZNF281 (4.6-fold, FDR = 0.04, *Supplementary file 3c*). Notably, ZNF281 is also one of the TFs whose predicted differential binding is most tightly linked with differential expression (*Figure 3—figure supplement 2d,e*). Overall, these data support a model whereby variants in ZNF281 motifs might have modulated ZNF281 binding in NPCs, thereby contributing to neural expression differences between modern and archaic humans.

## Potential phenotypic consequences of differential expression

In an attempt to assess the functional effects of the differential transcriptional activity we detected, we first sought to link each sequence to the gene(s) it might regulate in its endogenous genomic location. While most regulatory sequences are known to affect their closest gene (*Gasperini et al., 2019*; *Fulco et al., 2019*), some exert their function through interactions with more distal genes, often reflected in chromatin conformation capture assays, such as Hi-C interactions (*Gasperini et al., 2020*), or eQTL associations (*Gasperini et al., 2020*; *Jung et al., 2019*). To predict the genes linked to each sequence, we combined data from four sources: (1) proximity to TSS; (2) proximity to eQTLs (*GTEx Consortium, 2015*); (3) proximity to putative enhancers (*Fishilevich et al., 2017*); and (4) spatial interaction with promoters using Hi-C data (*Jung et al., 2019*) (see 'Materials and methods'). Using these data, we generated for each cell type a list of genes potentially regulated by each sequence. Overall, 1341 (75%) out of the 1791 active sequences were linked to at least one putative target gene (*Supplementary file 1h*).

To study the potential functional effects of differentially active sequences, we analyzed functions associated with their linked genes. To control for confounders such as cell type-specific regulation, gene length, and GC content, we compared differentially active sequences to other active sequences (instead of the genomic background), which minimizes inherent biases in the active sequences. First, we tested Gene Ontology terms and found an enrichment of the following terms within downregulating sequences: *vesicle-mediated transport* (6.6-fold, FDR = $1.9 \times 10^{-3}$, in osteoblasts),

*regulation of apoptotic process* (6.0-fold, FDR = $1.9\times10^{-3}$, in ESCs), *protein ubiquitination* (4.7-fold, FDR = $1.9\times10^{-3}$, in ESCs), multicellular organism development (3.3-fold, FDR = 0.01, in ESCs), and *protein transport* (3.3-fold, FDR = 0.02, in osteoblasts, *Figure 3—figure supplement 2f*, *Supplementary file 4a*). No enriched terms were found within upregulating sequences. To obtain a more detailed picture of phenotypic function, we ran Gene ORGANizer, a tool that uses monogenic disorders to link genes to the organs they affect (*Gokhman et al., 2017*). We analyzed the genes linked to differentially active sequences and found that for genes linked to sequences driving upregulation, the most enriched body parts belong to the vocal tract, that is, the vocal cords (5.0-fold, FDR = $1.3\times10^{-3}$), voice box (larynx, 3.8-fold, FDR = $4.8\times10^{-3}$), and pharynx (3.3-fold, FDR = $9.5\times10^{-3}$, all within ESCs, *Figure 4a*). Interestingly, we have previously reported that the most extensive DNA methylation changes in modern compared to archaic humans arose in genes affecting the vocal cords and voice box (*Gokhman et al., 2020*). Conversely, within sequences driving downregulation, the enriched body part is the cerebellum (3.0-fold, FDR = $9.2\times10^{-3}$, in NPCs, *Figure 4a*, *Supplementary file 4b*). This is in line with previous reports of cerebellar anatomy differences between modern humans and Neanderthals (*Neubauer et al., 2018*; *Gunz et al., 2019*; *Aiello and Dean, 2002*), including results suggesting that the biggest differences in brain anatomy are in the cerebellum (*Kochiyama et al., 2018*). These data also provide leads into the functional divergence of organs, like the voice box, that are not preserved in the fossil record.

Next, we delved into individual phenotypes associated with the differentially active sequences. To this end, we used the Human Phenotype Ontology (HPO) database (*Köhler et al., 2014*), a curated database of genes and the phenotypes they underlie in monogenic disorders. HPO covers a broad range of phenotypes related to anatomy, physiology, and behavior. We found that enriched phenotypes were involved in speech, heart morphology testicular descent, and kidney function (FDR <0.05, *Figure 4b*, *Supplementary file 4b*). These results reveal body parts and phenotypes that were particularly associated with gene expression changes between modern and archaic humans, and could be new candidates for phenotypes under selection.

## Downregulation of *SATB2* potentially underlies brain and skeletal differences

This catalog of *cis*-regulatory changes allows us to explore specific sequence changes that potentially underlie divergent phenotypes observed from fossils. To use the most robust data from lentiMPRA, we examined the 10 sequences that are differentially active across all three cell types and looked at their linked genes. To investigate the phenotypes that are potentially linked to these genes, we looked for those genes whose phenotypes can be compared to the fossil record (i.e., skeletal phenotypes). The only gene that fits these criteria was *SATB2*, a regulator of brain and skeletal phenotypes (*Zarate and Fish, 2017*). First, we analyzed its linked variant (C to T transition), which is at a position that is relatively conserved in vertebrates (GRCh38: 199,469,203 on chromosome 2, PhyloP score = 0.996). This position is found within a CpG island between two alternative TSS of *SATB2* (*Figure 4c*). It is also found in the first intron of *SATB2-AS1*, an antisense lncRNA which upregulates SATB2 protein levels (*Liu et al., 2017*). To determine if this position lies within a regulatory region, we investigated it for chromatin marks in modern humans. We found that it overlaps a DNase I-hypersensitive site (*ENCODE Project Consortium, 2012*) and shows many peaks of active histone modification marks in all three cell types (*Figure 4c*, *Supplementary file 1f*). Indeed, this sequence drives high expression in all three cell types (4th, 8th, and 19th percentile among active sequences, in ESCs, osteoblasts, and NPCs, respectively, FDR $<10^{-5}$ across all). Although this sequence shows hallmarks of activity in modern humans, compared to the archaic sequence, the modern human sequence is downregulating in all three cell types (−9% in ESCs, FDR = $6.8\times10^{-4}$, −27% in osteoblasts, FDR = $2.2\times10^{-42}$, and −12% in NPCs, FDR = $1.1\times10^{-7}$, *Figure 4d*). These results suggest that the ancestral version of this sequence possibly promoted even higher expression in archaic humans.

*SATB2* encodes a TF expressed in developing bone and brain. Its activity promotes bone formation, jaw patterning, cortical upper layer neuron specification, and tumorigenesis (*Zarate and Fish, 2017*). Genome-wide association studies show that common variants near and within *SATB2* are mainly associated with brain and bone phenotypes, such as reaction time, anxiety, mathematical abilities, schizophrenia, autism, bone density, and facial morphology (*Buniello et al., 2019*; *Claes et al., 2014*). Heterozygous loss-of-function (LOF) mutations in *SATB2* result in the *SATB2-*

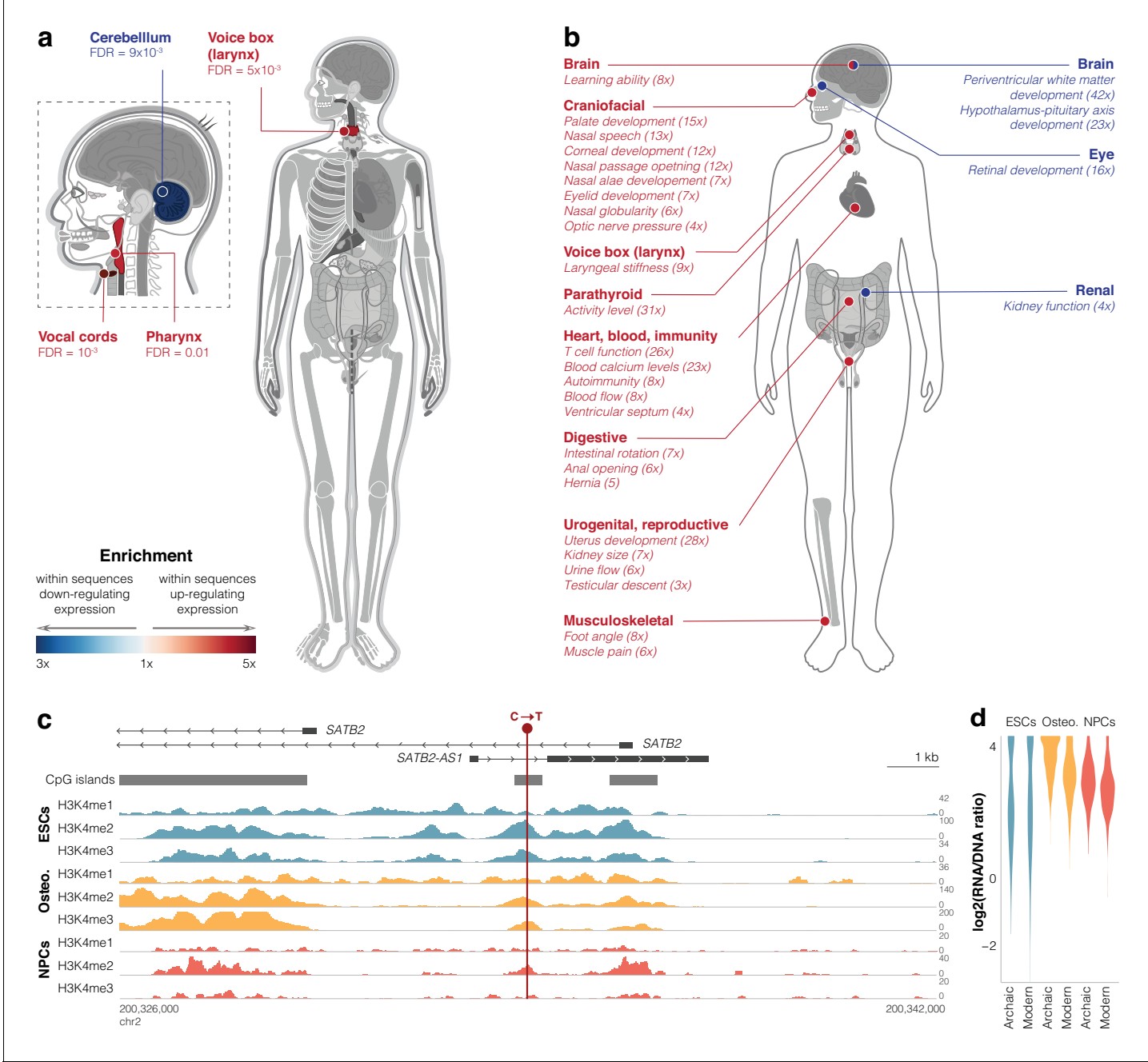

**Figure 4.** Differentially active sequences are linked to genes affecting the vocal tract and brain. (a) Gene ORGANizer enrichment map showing body parts that are significantly over-represented within genes linked to differentially active sequences (false discovery rate [FDR] <0.05). Organs are colored according to the enrichment scale. See *Supplementary file 4* for cell types. (b) Human Phenotype Ontology (HPO) phenotypes significantly enriched (FDR <0.05) within differentially active sequences. Fold enrichment is shown in parentheses. See *Supplementary file 4* for cell types. (c) CpG islands and read density of active histone modification marks (*Kundaje et al., 2015*) around the differentially active sequence in *SATB2* (GRCh37 genome version). (d) Violin plots of archaic vs. modern activity of the differentially active sequence in *SATB2*.

associated syndrome, which primarily affects neurological and craniofacial phenotypes. This includes speech delay, behavioral anomalies (e.g., jovial personality, aggressive outbursts, and hyperactivity), autistic tendencies, small jaws, dental abnormalities, and morphological changes to the palate (*Zarate et al., 1993*). Additionally, reduced functional levels of SATB2 due to heterozygous LOF have been shown to be the cause of these phenotypes in both human (*Zarate and Fish, 2017*; *Zarate et al., 1993*; *Gigek et al., 2015*; *Qian et al., 2019*) and mouse (*Li et al., 2017*; *Zhang et al.,

*2019*; *Dobreva et al., 2006*). Because these phenotypes are driven by changes to functional SATB2 levels (*Zarate and Fish, 2017*), we conjectured that the differential expression of *SATB2* predicted from lentiMPRA might be linked to divergent modern human phenotypes. Thus, we examined whether the phenotypes *SATB2* affects are divergent between archaic and modern humans (e.g., if modern human jaw size is different than the jaw size of archaic humans). We focused on phenotypes available for examination from the fossil record, primarily skeletal differences between modern humans and Neanderthals. From HPO, we generated a list of 17 phenotypes known to be affected by *SATB2* and found that 88% (15) of them are divergent between these human groups (*Supplementary file 5*). These include the length of the skull, size of the jaws, and length of the dental arch. Next, based on *SATB2* downregulation in modern humans predicted from lentiMPRA, we examined whether the direction of a phenotypic change between patients and healthy individuals matches the direction of phenotypic change between modern and archaic humans. For example, given that *SATB2*-associated syndrome patients have smaller jaws, we tested if modern human jaws are smaller compared to archaic humans. If *SATB2* expression is not in fact related to phenotypic divergence, there is a 50% likelihood for a given phenotype to match the fossil record. Yet, we observed a match in direction in 80% of the phenotypes (12 out of 15, *Supplementary file 5*). This includes smaller jaws, flatter face, and higher forehead in modern compared to archaic humans. Overall, the observed number of phenotypes that are both divergent and match in their direction of change is 2.3-fold higher than expected by chance (p=$1.3\times10^{-4}$, hypergeometric test, *Supplementary file 5*, see 'Materials and methods'). Together, these data support a model whereby the C→T substitution in the putative promoter of *SATB2*, which emerged and reached fixation in modern humans, possibly reduced the expression of *SATB2* and possibly affected brain and craniofacial phenotypes. However, further evidence is required to elucidate the potential role of this variant in modern human evolution.

## Discussion

Identifying noncoding sequence changes underlying human traits is one of the biggest challenges in genetics. This is particularly difficult in ancient samples, where regulatory information is scarce (*Yan and McCoy, 2020*; *Gokhman et al., 2016*). Here, we use an MPRA-based framework to study how sequence changes shaped human gene regulation. By comparing modern to archaic sequences, we investigated the regulatory potential of each of the 14,042 single-nucleotide variants that emerged and reached fixation or near fixation in modern humans. We found an association between divergent TF motifs and the sequences driving expression changes, suggesting that changes to TF binding might have played a central role in shaping divergent modern human expression. Our results also suggest that genes affecting the vocal tract and cerebellum might have been particularly affected by these expression changes, which is in line with previous comparisons based on the fossil record (*Neubauer et al., 2018*; *Gunz et al., 2019*; *Aiello and Dean, 2002*; *Kochiyama et al., 2018*) and DNA methylation (*Gokhman et al., 2020*). More importantly, these results provide candidate sequence changes underlying these evolutionary trends.

LentiMPRA is designed for linking DNA sequence changes to expression changes *en masse*. Notably, it has limitations that could influence our results, mainly by potentially generating false negatives. First, our lentiMPRA library inserts were limited to ~200 bp in length, due to oligonucleotide synthesis technical restrictions, which may be insufficient to detect the activity of longer enhancer sequences (*Inoue et al., 2017*). Second, some minimally active sequences may not be expressed at a high enough level to pass our limit of detection. At the same time, some minimally active sequences may not be biologically significant. Third, some sequences may regulate expression post-transcriptionally, which lentiMPRA is not designed to detect. Fourth, since test sequences are randomly integrated into the genome, sequences that are dependent on their endogenous genomic environments (e.g., on nearby TF binding sites) might show reduced activity when inserted in new locations, while others might show activity that they otherwise would not have. Our design partially addresses this through the use of antirepressors and multiple independent integrations, which are intended to dilute location-specific effects. Additionally, all biases are expected to similarly affect the modern and archaic human versions of each sequence (*Inoue et al., 2017*). Fifth, transcriptional repression is less likely to be detected due to the low basal activity of the mP used. Sixth, the level of sequence activity may depend on more than one variant (including non-fixed variants, which we have not

tested here). In the cases of non-fixed variants, the extent of differential activity could vary between individuals. At the same time, in the 10% of sequences that include more than one fixed variant, it is generally impossible to determine which of the variants drives the differential activity (with the exception of cases with more than two variants where the tiled sequences include a different combination of these variants).

Finally, differences in the *trans* environment of a cell could have an effect on the ability of a sequence to exert its *cis*-regulatory effect, resulting in cell type-specific *cis*-regulatory effects, as we observed in our data. The *trans* environment of the same cell type might also differ between two organisms. However, the majority of the *cis*-regulatory changes we observed would be expected to be present in archaic human cells as well, considering that such conservation has been observed between substantially more divergent organisms (e.g., human-chimpanzee [*Ryu et al., 2018*] and human-mouse [*Mattioli et al., 2020*]). In other words, while *trans*-regulatory changes play a key role in species divergence, the *trans* environments of the same cell type in two closely related organisms tend to affect *cis*-regulation similarly. Despite these caveats, MPRAs have been repeatedly shown to be able to replicate the activity of sequences in their endogenous context (*Inoue et al., 2017*; *Klein et al., 2020*; *Kircher et al., 2019*).

Importantly, when genomes from additional modern human individuals are sequenced and new variants mapped, it might become clear that some of the variants we analyzed have not reached fixation. However, regardless of whether they are completely fixed or not, these variants represent derived substitutions that likely emerged in modern humans and spread to considerable frequency. Further investigation is required to determine when they emerged, how rapidly they spread, and whether their effect was neutral or adaptive.

As expected, we observed differences in activity and differential activity between cell types (*Tewhey et al., 2016*; *Kircher et al., 2019*; *Mattioli et al., 2020*). Although some of this variation is likely biological (i.e., cell type-specific gene regulation), it is difficult to determine what proportion of it is due to biological vs. technical factors (e.g., differences in lentivirus preparation, infection rate, or cell growth, see 'Materials and methods'). Importantly, these differences are expected to result in false negatives rather than false positives. In other words, some of the sequences that appear as active or differentially active in one cell type might actually be active or differentially active in additional cell types (including cell types that were not tested in this study). Thus, we largely refrained from comparisons between cell types and the overlap observed in *Figure 2a* and *Figure 3a* should not be used to define such similarities. Rather, these diagrams should be used to examine the replicability of our results. Despite these caveats and limitations, lentiMPRA is a powerful high-throughput tool to characterize the regulatory activity of derived variants, and indeed has become a common assay to study the capability of sequences to promote expression (*Chatterjee and Ahituv, 2017*).

With this method, we found that 1791 (13%) of the 14,042 sequence pairs can promote expression in at least one of the three cell types tested, and that 405 (23%) of these active sequences show differential activity between modern and archaic humans (average fold-change: 1.24×, standard deviation 0.18, *Figure 2—figure supplement 1a–c*). Interpreting these results in light of previous MPRAs is challenging, not only because of key differences in statistical power and experimental design (e.g., sequence length), but also because of differing variant selection processes for each MPRA. With the exception of highly repetitive regions, which were removed from our library for technical reasons, the sequences we selected included all known modern human-derived fixed or nearly fixed variants (see 'Materials and methods'). Conversely, previous reporter assays and MPRAs on human intra- or inter-species variation used biased sets of variants by selecting sequences with putative regulatory function (e.g., eQTLs [*Tewhey et al., 2016*], TF binding sites [*Weyer and Pääbo, 2016*], ChIP-seq peaks [*Klein et al., 2018*], or TSS [*Mattioli et al., 2020*]) and/or regions showing particularly rapid evolution (e.g., human accelerated regions; *Ryu et al., 2018*; *Uebbing et al., 2021*; *Prabhakar et al., 2008*; *Capra et al., 2013*). In line with the fact that our data was not pre-filtered for putative regulatory regions, the proportion of active sequences we observed tends to be slightly lower than these previous studies. However, the magnitude of differential activity, as well as the fraction of differentially active sequences out of the active sequences, was similar to previous studies (*Weyer and Pääbo, 2016*; *Tewhey et al., 2016*; *Klein et al., 2018*; *Ryu et al., 2018*; *Uebbing et al., 2021*; *Mattioli et al., 2020*; *Prabhakar et al., 2008*; *Capra et al., 2013*). At the same time, we were capable of measuring regulatory activity in regions that would otherwise be

excluded by filtering for a specific set of marks. Thus, future MPRAs on unfiltered sets of variants will enable the comparison of the patterns we observed to patterns within modern humans, between more deeply divergent clades, and of non-fixed modern-archaic differences.

Our results also suggest that differentially active sequences are over-represented within putative enhancers in NPCs (*Figure 1—figure supplement 1c–d*, *Supplementary file 2*). Enhancers have been suggested to be an ideal substrate for evolution because of their tissue specificity and temporal modularity (*True and Carroll, 2002*). Indeed, previous studies of introgression between archaic and modern humans suggested that enhancers are some of the most divergent regions between modern and archaic humans (*Petr et al., 2019*; *Silvert et al., 2019*; *Telis et al., 2020*). In line with the enrichment we observed in NPCs, brain-related putative enhancers show particularly low introgression, perhaps suggesting that the modern human sequences in these regions were adaptive (*Silvert et al., 2019*; *Telis et al., 2020*). To fully characterize the underlying mechanisms of differential activity in enhancers, it is important to disentangle the various factors and confounders that might contribute to this enrichment. There are several alternative explanations for the enrichment we observe, namely that variants within enhancers could be more likely to alter expression compared to other active sequences, or they could be particularly detectable in lentiMPRA. This could be tested using saturation mutagenesis MPRAs (*Kircher et al., 2019*) to compare the effect of random mutations in enhancer and non-enhancer modern human-derived active sequences.

Our results suggest that differentially active sequences are not randomly distributed across the genome, but rather tend to be linked to genes affecting particular body parts and phenotypes. The most pronounced enrichment was in the vocal tract, that is, the vocal cords, larynx, and pharynx. This was evident in the Gene ORGANizer analysis, where these organs are over-represented by up to fivefold, as well as in the HPO analysis, where some of the most enriched phenotypes are *nasal speech*, *palate development*, *nasal passage opening*, and *laryngeal stiffness* (*Figure 4b*, *Supplementary file 4c*). Overall, 53 of the 407 differentially active sequences were linked to genes which are known to affect one or more vocal tract phenotypes. Previous reports have also suggested that the vocal tract went through particularly extensive regulatory changes between modern and archaic humans (*Gokhman et al., 2020*), as well as between humans and chimpanzees (*Gokhman et al., 2021*; *Prescott et al., 2015*). Intriguingly, the anatomy of the vocal tract differs between humans and chimpanzees and has been suggested to affect human phonetic range (*Lieberman, 2007*). Comparing the anatomy of archaic and modern human larynges is currently impossible because the soft tissues of the larynx rapidly decay postmortem. However, together with these previous reports (*Gokhman et al., 2020*; *Gokhman et al., 2021*; *Prescott et al., 2015*), our results enable the study of vocal tract evolution from a genetic point of view and suggest that genes influencing the modern human vocal tract have possibly gone through regulatory changes that are not shared by archaic humans.

We also identified an enrichment of brain-related phenotypes, particularly those affecting the size of the cerebellum (*Figure 4*, *Supplementary file 4b,c*). The cerebellum is involved in motor control and perception, as well as more complex functions such as cognitive processing, emotional regulation, language, and working memory (*Mariën et al., 2014*). Interestingly, the cerebellum has been described as the most morphologically divergent brain region between modern and archaic humans (*Neubauer et al., 2018*; *Kochiyama et al., 2018*). Evidence of divergent brain and cerebellar evolution can also be found at the regulatory level. Studies of Neanderthal alleles introduced into modern humans through introgression provide a clue as to the functional effects of divergent loci between archaic and modern humans. These works have shown that many of the introgressed sequences were likely negatively selected, with the strongest effect in regulatory regions (*Petr et al., 2019*; *Silvert et al., 2019*), particularly in brain enhancers (*Telis et al., 2020*). Studies of introgressed sequences have also shown that the cerebellum is one of the regions with the most divergent expression between Neanderthal and modern human alleles (*McCoy et al., 2017*). Together with our results, these data collectively suggest that sequences separating archaic and modern humans are particularly linked to functions of the brain, and especially the cerebellum.

Functional information on archaic human genomes is particularly challenging to obtain because of the postmortem decay of RNA and epigenetic marks in ancient samples. MPRA not only provides a new avenue to identify differential regulation in archaic samples, but also reveals the sequence changes underlying these differences. Here, we present a catalog providing regulatory insight into the sequence changes that separate modern from archaic humans. This resource will hopefully help

assign functional context to various signatures of sequence divergence, such as selective sweeps and introgression deserts, and facilitate the study of modern human evolution through the lens of gene regulation.

## Materials and methods

### Code and data availability

Code is available for download on Github: https://github.com/weiss19/AH-v-MH; *Weiss, 2021*; copy archived at swh:1:rev:a75b6f0b7d278cb0388e52b3d491e262be77c206. Data was deposited in GEO under accession number: GSE152404.

### Selection of fixed, derived variants and design of DNA oligonucleotides

We selected the variants for our lentiMPRA in the following manner. As a basis, we used the list of 321,820 modern human-derived single-nucleotide changes reported to differ between modern humans and the Altai Neanderthal genome (*Prüfer et al., 2014*). We then filtered this list to include only positions where the Vindija Neanderthal (*Prüfer et al., 2017*) and Denisovan sequences (*Meyer et al., 2012*) both match the Altai Neanderthal variant, and are also not polymorphic in any of the four ape species examined (61 *Pan troglodytes,* 10 *Pan paniscus,* 15 *Gorilla beringei,* and 28 *Gorilla gorilla*) (*de Manuel et al., 2016*). Next, we excluded loci which had any observed variation within modern humans in dbSNP, as annotated by *Prüfer et al., 2014*, or in the 1000 Genomes Project (phase 3) (*Auton et al., 2015*). Finally, for technical limitations in downstream synthesis and cloning, we excluded variants at which the surrounding 200 bp had >25% repetitive elements as defined by RepeatMasker (*Smit et al., 1996*). The resulting list contained 14,297 sequences and was used to design the initial set of DNA fragments. Upon completion of the lentiMPRA, another high-coverage Neanderthal genome (the Chagyrskaya Neanderthal) was published (*Mafessoni et al., 2020*), and we subsequently also filtered out loci at which the Chagyrskaya Neanderthal genome did not match the ancestral sequence, bringing the final list of analyzed loci to 14,042 (28,082 archaic and modern sequences, *Supplementary file 1a-c*).

We designed DNA fragments (oligonucleotides, hereinafter oligos) centered on each variant, including the 99 bp upstream and 100 bp downstream of each variant (200 bp total). For each variant we designed two fragments, one with the ancestral (archaic human and ape) sequence and one with the derived (modern human) sequence. For cases where two or more variants would be included in the same oligo, we used either derived-only (modern human) or ancestral-only (archaic human and ape) variants throughout the oligo. The average variants per oligo out of the 14,042 oligos was 1.1, with 12,680 containing one variant, 1259 containing two, 96 containing three, and 7 containing four. We also included 100 negative control fragments, created by randomly picking 100 of the designed DNA fragments and scrambling their sequence (*Supplementary file 1e*). Lastly, we incorporated 299 positive control fragments (*Ryu et al., 2018*; *Prabhakar et al., 2008*; *Visel et al., 2007*; *Inoue et al., 2019*; *Hojo et al., 2016*; *Meyer et al., 2016*; *Khalid et al., 2018a*; *Khalid et al., 2018b*; *Loots et al., 2005*; *Fukami et al., 2006*; *Kawane et al., 2018*) (i.e., expected to drive expression; *Supplementary file 1d*). As the library was infected into three cell types (see later), we designed positive controls for each of the cell types. For human ESCs and human NPCs, we used sequences which were previously shown to drive expression in MPRA in each of these cell types (*Supplementary file 1d*). For fetal osteoblast cells (Hobs), we used putative and confirmed enhancers from mouse and human (*Supplementary file 1d*). 15 bp adapter sequences for downstream cloning were added to the 5' (5'-AGGACCGGATCAACT) and 3' (5'-CATTGCGTGAACCGA) ends of each fragment, bringing the total length of each fragment to 230 bp. We synthesized each fragment as an oligonucleotide through Agilent Technologies, twice independently to minimize synthesis errors (*Supplementary file 1i*).

### Production of the plasmid lentiMPRA library and barcode association sequencing

The plasmid lentiMPRA library was generated as described in *Gordon et al., 2020*. In brief, the two independently synthesized Agilent Technologies oligo pools were amplified separately via a five-cycle PCR using a different pairs of primers for each pool (forward primers, 5BC-AG-f01.1 and 5BC-

AG-f01.2; reverse primers, 5BC-AG-r01.1 and 5BC-AG-r01.2; *Supplementary file 1i*), adding an mP downstream of the test sequence. A second round of five-cycle PCR was performed with the same primers for both pools (5BC-AG-f02 and 5BC-AG-r02; *Supplementary file 1i*) to add a 15 bp random barcode downstream of the mP. The two pools were then combined at a 1:1 ratio and cloned into a doubled digested (AgeI/SbfI) pLS-SceI vector (Addgene, 137725) with NEBuilder HiFi Master Mix (NEB). The resulting plasmid lentiMPRA library was electroporated into 10-beta competent cells (NEB) using a Gemini X2 electroporation system (BTX) (2 kv, 25 µF, 200 Ω) and allowed to grow up overnight on twelve 15 cm 100 mg/mL carbenicillin LB agar plates. Colonies were pooled and midi-prepped (Qiagen). We collected approximately 6 million colonies, such that ~200 barcodes were associated with each oligo on average. To determine the sequences of the random barcodes and which oligos they were associated with, we first amplified a fragment containing the oligo, mP, and barcode from each plasmid in the lentiMPRA library using primers that contain Illumina flow cell adapters (P7-pLSmp-ass-gfp and P5-pLSmP-ass-i#, *Supplementary file 1i*). We sequenced these amplified sequences with a NextSeq 150PE kit using custom primers (R1, pLSmP-ass-seq-R1; R2 [index read], pLSmP-ass-seq-ind1; R3, pLSmP-ass-seq-R2, *Supplementary file 1i*) to obtain approximately 150 million total reads. We later did a second round of barcode association sequencing of these fragments to obtain approximately 76 million additional reads, for a combined total of 225,592,667 reads. To associate barcodes with oligos, we first mapped read pairs (R1 and R3) to the original list of 28,993 oligos using bowtie2 (–very-sensitive) (*Langmead and Salzberg, 2012*). Next, we filtered out pairs of reads that (1) did not map to the same oligo, (2) did not have at least one of the reads in the pair with a mapping quality of ≥6, or (3) did not have the 'proper pair' SAM designation. We linked each pair of reads with the read covering its barcode (R2) and saved only those barcode reads having at least a quality score of 30 across all 15 bases in the R2 read. We removed any barcodes associated with more than a single unique oligo (i.e., 'promiscuous' barcodes), as well as any barcodes where we did not see evidence of its oligo association at least three times. We then created a list of barcode-oligo associations – this final list comprised 3,495,698 unique barcodes spanning 28,678 oligos (98.9% of the original list of 14,297 variant sequence pairs, 100 negative sequences and 299 positive control sequences), which we refer to as the barcode-oligo association list.

## Cell culture and differentiation

Human fetal osteoblasts were purchased from Cell Applications Inc (406 K-05f, tested negative for mycoplasma) and were maintained in Osteoblast Growth Medium (Cell Applications Inc). For passaging, cells were washed with 1× PBS, dissociated with trypsin/EDTA (Cell Applications Inc), and plated at approximately 5000 cells/cm$^2$. H1-ESCs (ESCs, WiCell WA-01, RRID:CVCL_9771, identity authenticated via STR profiling, and tested negative for mycoplasma) were cultured on Matrigel (Corning) in mTeSR1 media (STEMCELL Technologies) and medium was changed daily. For passaging, cells were dissociated using StemPro Accutase (Thermo Fisher Scientific), washed and re-plated on Matrigel-coated dishes at a dilution of 1:5 to 1:10 in mTeSR1 media supplemented with 10 µM Y-27632 (Selleck Chemicals). ESCs were differentiated into NPCs by dual-Smad inhibition as previously described (*Chambers et al., 2009*; *Inoue et al., 2019*). Briefly, ESCs were cultured in mTeSR1 media until the cells became 80% confluent and then the media was replaced with neural differentiation media consisting of KnockOut DMEM (Life Technologies) supplemented with KnockOut Serum Replacement (Life Technologies), 2 mM L-glutamine, 1× MEM-NEAA (Life Technologies), 1× beta-mercaptoethanol (Life Technologies), 200 ng/mL Recombinant mouse Noggin (R and D systems), and 10 µM SB431542 (EMD Millipore). On day 4 of differentiation, the neural differentiation media was gradually replaced by N2 media (DMEM/F12 [Thermo Fisher Scientific] supplemented with N2 [Thermo Fisher Scientific]) every 2 days (3:1 ratio on day 6, 1:1 on day 8, and 1:3 on day 10) while maintaining 200 ng/mL Noggin and 10 µM SB431542. On day 12, cells were dissociated into single-cell using TrypLE Express (Thermo Fisher Scientific) and cultured in N2B27 media (1:1 mixture of N2 media and Neurobasal media [Thermo Fisher Scientific] with B27 [Thermo Fisher Scientific] supplemented with 20 ng/mL bFGF [R and D systems] and 20 ng/mL EGF [Millipore sigma]) on Matrigel-coated dish. NPCs were maintained in N2B27 with bFGF and EGF for a month and used for the following experiments at passage 15.

NPCs were validated through RT-qPCR at passage 1 (after 1 week of culturing in N2B27 media supplemented with bFGF and EGF) and at passage 10. RT-qPCR primers were designed for neural

marker genes: *SOX1/2*, *NES* (*NESTIN*), *MAP2*; glial marker genes: *GFAP*, *OLIG2*; mesoderm marker genes: *T*(*BRA*), *GSC*; and endoderm marker genes: *SOX17*, *FOXA2* (*Supplementary file 1j*). Expression of each marker was compared to *HPRT* expression (*Supplementary file 1h*). Additionally, validation via RNA-seq at passage 1 was performed. Results can be found in Figure 7A and D of *Inoue et al., 2019* (data in GEO under accession number: GSE115046).

## Cell line infection with lentiMPRA library, RNA- and DNA-seq, and read processing

Lentivirus was produced and packaged with the plasmid lentiMPRA library in twelve 15 cm dishes of HEK293T cells using the Lenti-Pac HIV expression packaging kit, following the manufacturer's protocol (GeneCopoeia). Additional lentivirus was produced as needed in batches of ten 15 cm dishes. Lentivirus containing the lentiMPRA library (referred to hereafter as lentivirus) was filtered through a 0.45 μm PES filter system (Thermo Fisher Scientific) and concentrated with Lenti-X concentrator (Takara Bio). Titration reactions using varying amounts of lentivirus were conducted on each cell type to determine the best volume to add, based on an optimal number of viral particles per cell, as described in *Gordon et al., 2020*. Lentiviral infection, DNA/RNA extraction, and barcode sequencing were all performed as described in *Gordon et al., 2020*. Briefly, each replicate consisted of approximately 9.6 million cells each of ESC and osteoblast, and 20 million cells of NPC. ESC and osteoblast cells were seeded into four 10 cm dishes per replicate (with approximately 2.4 million cells in each dish), while NPCs were seeded into five 10 cm dishes per replicate (with approximately 4 million cells per dish). Additional cells were used for NPCs due to decreased efficiency of DNA/RNA extraction in NPCs. Three replicates were performed per cell type. Cells were infected with the lentiMPRA library at an MOI of 50 for NPCs and osteoblasts, and an MOI of 10 for ESCs. We used a lower MOI for ESC because the cells are very sensitive to infection and an MOI higher than 10 would result in cell death. For ESC and osteoblasts, cell media was changed to include 8 μg/mL polybrene before the addition of the lentiMPRA library to increase infection efficiency. The media was replaced with growth media without polybrene approximately 24 hr after infection. Infected cells were grown for 3 days before combining the plates of each replicate for extraction of RNA and DNA via the Qiagen AllPrep mini kit (Qiagen). We subsequently purified mRNA from the RNA using the Oligotex mRNA prep kit (Qiagen) and synthesized cDNA from the resulting mRNA with SuperScript II RT (Invitrogen), using a primer containing a unique molecular identifier (UMI) (P7-pLSmp-ass16UMI-gfp, *Supplementary file 1i*). DNA fragments were amplified from both the isolated DNA and generated cDNA, keeping each replicate and DNA type separate, with three-cycle PCR using primers that include adapters necessary for sequencing (P7-pLSmp-ass16UMI-gfp and P5-pLSmP-5bc-i#, *Supplementary file 1i*). These primers also contained a sample index for demultiplexing and a UMI for consolidating replicate molecules (see later). A second round of PCR was performed to amplify the library for sequencing using primers targeting the adapters (P5, P7, *Supplementary file 1i*). The fragments were purified and further sequenced with six runs of NextSeq 15PE with 10-cycle dual index reads, using custom primers (R1, pLSmP-ass-seq-ind1; R2 [read for UMI], pLSmP-UMI-seq; R3, pLSmP-bc-seq; R4 [read for sample index], pLSmP-5bc-seq-R2, *Supplementary file 1i*). Later, an additional two runs of 15PE of only the ESC samples were performed due to lower lentivirus infection efficiency in this cell type. Each sample's R1 and R3 reads (containing the barcode) were mapped with bowtie2 (*Langmead and Salzberg, 2012*) (–very-sensitive) to the barcode-oligo association list. Next, we applied several quality filters on the resulting alignments. We first filtered out read pairs that did not map as proper pairs, and then ensured the mapped sequence completely matched the known barcode sequence by requiring that both R1 and R3 reads have CIGAR strings = 15M, MD flags = 15, and a mapping quality of at least 20. Next, we consolidated read abundance per barcode by selecting only reads with unique UMIs, the result being abundance counts for each barcode, across each replicate library of each cell type for both RNA and DNA.

Data was deposited in GEO under accession number GSE152404.

## Measurement of expression and differential expression

We used the R package MPRAnalyze (*Ashuach et al., 2019*) (version 1.3.1, https://github.com/YosefLab/MPRAnalyze) to analyze lentiMPRA data. To determine which oligos were capable of promoting expression, we modeled replicate information into both the RNA and DNA models of

MPRAnalyze's quantification framework (rnaDesign = ~ replicate and dnaDesign = ~ replicate) and extracted alpha, the transcription rate, for each oligo. MPRAnalyze used the expression of our 100 scrambled oligos as a baseline against which to measure the level of expression of each tested oligo. We corrected the mean absolute deviation (MAD) score-based p-values from MPRAnalyze for multiple testing across tested oligos, including positive controls and excluding scrambled sequences, using the Benjamini-Hochberg method, thus generating an MAD score-based expression FDR for each oligo. For each variant and for each cell type, we looked at both the archaic and modern sequence oligos and assigned an oligo as potentially capable of driving expression if it had an FDR $\leq$ 0.05 in at least one sequence, and at least 10 barcodes in both sequences (*Supplementary file 1a-c*). This left 2097 sequences in ESCs, 1059 in osteoblasts, and 664 in NPCs. Next, we applied a second test for activity, to account for potential overestimation of active sequences in ESCs due to the lower lentiviral infection efficiency in these cells. We aggregated UMI-normalized read abundances across all barcodes of each oligo, across all replicates in a given cell type, and calculated a simple ratio of expression as RNA abundance normalized to DNA abundance (RNA/DNA ratio). Next, similarly to *Kwasnieski et al., 2014*, we determined an RNA/DNA ratio threshold per cell type. This was done by first removing scrambled sequences that show RNA/DNA ratios >2 standard deviations away from the average RNA/DNA ratio of all of the scrambled sequences, as these likely represent oligos that are, by chance, capable of driving some expression. This left 95 scrambled sequences in ESCs, 94 in osteoblasts, and 97 in NPCs. Then, we used the distribution of RNA/DNA ratios of the remaining scrambled sequences to assign an FDR for each of the non-scrambled oligos. FDR was calculated as the fraction of scrambled sequences that showed an RNA/DNA ratio as high or higher than each non-scrambled oligo. Only oligos that passed both tests described above (FDR $\leq$0.05 in each test) were considered as 'active' (i.e., capable of driving expression). This resulted in 1183 sequences in ESCs, 814 in osteoblasts, and 602 in NPCs.

To measure differential expression between archaic and modern sequences, we used MPRAnalyze's comparative framework. In essence, this tool uses a barcode's RNA reads as an indicator of expression level and normalizes this to the DNA reads as a measure of the number of genomic insertions of that barcode (i.e., the number of fragments from which RNA can be transcribed). MPRAnalyze uses information across all the barcodes for both alleles of a given sequence, as well as information across all replicates. For the terms of the model, we included replicate information in the RNA, DNA, and reduced (null) models, allele information in the RNA and DNA models, and barcode information only in the DNA model (rnaDesign = ~ replicate + allele, dnaDesign = ~ replicate + barcode + allele, reducedDesign = ~ replicate). We extracted p-values and the differential expression estimate (fold-change of the modern relative to archaic sequence). Then, we corrected the p-values of the set of active oligos (see above) for multiple testing with the Benjamini-Hochberg method to generate an FDR for each sequence. We set a cutoff of FDR $\leq$ 0.05 to call a sequence capable of driving differential expression. From this we generated, for each cell type, a list of sequences with differential expression between the archaic and modern alleles (*Supplementary file 1a-c*).

We tested agreement between replicates by examining how many differentially active sequences show disagreement between the three replicates in the direction of their differential activity. We found that our dataset shows high between-replicate agreement, with the majority of sequences showing the same directionality across all three replicates (ESCs: 76%, osteoblasts: 78%, NPCs: 86%, compared to 25% expected by chance, $p<10^{-16}$ for all three cell types, one-tailed Binomial test, *Supplementary file 1k*). Importantly, the $\log_2$(fold-change) (LFC) of the disagreeing replicate tends to cross the 0 line only marginally: the median LFC of the disagreeing replicate is 0.05 compared to 0.3 in the agreeing replicates. We also tested activity levels and found no evidence of lower activity in sequences with disagreement (p=0.27, one-tailed *t*-test). However, their absolute LFC tends to be slightly lower (0.25 vs. 0.32, $p=6\times10^{-5}$, one-tailed *t*-test).

## Luciferase validation assays

Each assayed oligo was synthesized by Twist Biosciences and cloned into the pLS-mP-Luc vector (Addgene 106253) upstream of the luciferase gene. Lentivirus was generated independently for each vector using techniques as described for MPRA (see above), with the omission of the filtering and concentration step, which was replaced with the collection of the entirety of the cell culture media for use in subsequent infections. In addition, pLS-SV40-mP-Rluc (Addgene 106292), to adjust

for infection efficiency, was added at a 1:3 ratio to the assayed vector for a total of 4 µg for lentivirus production. We infected each cell type individually with each viral prep. The amount of lentivirus added was based on titrations in which varying amounts of a subset of viral preps were added to each cell type and cell death was observed 3 days post infection; the virus volume that produced between 30% and 50% death was used for subsequent experiments. Approximately 20,000 cells were plated in 96-well plates and grown for 24–48 hr (~70% confluent) before the addition of lentivirus. For osteoblasts and ESCs, 8 µg/mL polybrene was added to the culture media at the same time as the addition of the lentivirus. The media was changed 24 hr after infection and cells were grown for an additional 48 hr. The cells were then washed with PBS and lysed. Firefly and renilla luciferase expression were measured using the Dual-Luciferase Reporter Assay System (Promega) on the Glo-Max plate reader (Promega). Each oligo was tested using two biological replicates on different days and each biological replicate consisted of three technical replicates. Activity of a given oligo was calculated by normalizing the firefly luciferase activity to the renilla luciferase. We then calculated the LFC between the modern and archaic alleles as $\log_2$(modern/archaic). A full list of oligos tested and their LFC can be found in *Supplementary file 1a-c*.

We found that the mean difference in fold-change between replicates was threefold lower for the differentially active vs. other active sequences (0.22 vs. 0.60), and that the variance of these differences was ninefold lower for differentially active sequences compared to other active sequences (0.09 vs. 0.83, *Supplementary file 1k*), suggesting that differentially active sequences reflect a true biological signal.

## Predicting target genes

To connect the surrounding locus of each variant to genes it potentially regulates, we combined four data sources. For each locus, we generated four types of gene lists, based on four largely complementary approaches: (1) overlap with known eQTLs, (2) spatial interaction with promoters, (3) proximity to putative enhancers, and (4) proximity to a TSS (*Supplementary file 1h*). Each data source was obtained and incorporated into each type of list as described below.

### Proximity to known eQTLs

eQTLs are genetic variants between individuals shown to be associated with expression differences. We reasoned that the target genes of the sequence surrounding a variant are potentially similar to the target genes of nearby eQTLs. We downloaded eQTLs and their associated genes from GTEx (*GTEx Consortium, 2015*) (http://www.gtexportal.org, v8 on August 26, 2019) and overlapped the locations of each eQTL with our list of sequences. We linked the target genes of any eQTLs within ±1 kb to each variant. We used all tissue types reported by GTEx, for each cell type in the lentiM-PRA. Out of the 14,042 loci, 9503 were found within ±1 kb of an eQTL, with 83,777 eQTLS overall overlapping them.

### Spatial interaction with a promoter via Hi-C data

High-throughput chromosome conformation capture (Hi-C) techniques map spatial interactions between segments of DNA. We reasoned that if a variant is found within or near a region that was shown to interact physically with a promoter, that variant could be in a region involved in regulating that promoter. We downloaded promoter capture Hi-C data from *Jung et al., 2019*, containing a list of all the significant interactions between promoters and other segments of the genome across 27 tissue and cell types. We overlapped our variants with the locations of interacting genomic fragments to find interactions within ±10 kb of each variant. We then linked each variant with the promoters that each interacting fragment was shown to contact. We repeated this process twice: once to obtain a cell type-specific list and once to obtain a generic list. For the cell type-specific (stringent) list of locus-gene links, we included only those interactions observed in cell types corresponding to the cell lines used in our lentiMPRA: ESCs, NPCs, and mesenchymal stem cells (MSCs) as an approximation for osteoblasts (given that osteoblast Hi-C data is not publicly available to the best of our knowledge, and that osteoblasts differentiate from MSCs). For the generic (non-stringent) list, we used interactions across any of the 27 tissue and cell types analyzed by *Jung et al., 2019*. Out of the 14,042 loci, 4688 overlapped at least one region that interacts with a promoter.

### Putative enhancers

Lastly, we checked which of our variants were in previously reported putative enhancers. To this end, we downloaded the GeneHancer database (*Fishilevich et al., 2017*) V4_12 and searched for putative enhancers within ±10 kb of each of our variants, linking each variant to the target genes of each putative enhancer within that distance. GeneHancer provides 'elite' or 'non-elite' status to their defined enhancer-target gene connections depending on the strength of the evidence supporting each connection. Using this information, we repeated the process twice: once for the elite status and once for all annotations. Out of the 14,042 loci, 5017 overlapped at least one putative enhancer.

### Promoters

Promoters were defined as the region 5 kb upstream to 1 kb downstream of GENCODE (*Harrow et al., 2012*) v29 GRCh38 TSS. If a variant fell within this region, we linked it to that TSS's gene. Each variant was assigned to all the promoters it fell within. Out of the 14,042 loci, 1466 were found within a promoter.

Overall, 11,207 out of the 14,042 loci were linked to at least one putative target gene, with a median of four target genes per locus. Of the remaining loci, 2830 were linked to their closest TSS, regardless of distance. The last five without hg38 coordinates for their closest TSS were not linked to a gene. Importantly, these links do not necessarily mean that these target genes are regulated by these loci, but rather they serve as a list of potential target genes for the loci showing a regulatory function through lentiMPRA.

## DNA methylation in active and differentially active sequences

The four highest resolution DNA methylation maps for modern and archaic bone samples were taken from *Gokhman et al., 2014* and *Gokhman et al., 2020*. Promoter sequences were defined as sequences within 5 kb upstream to 1 kb downstream of a TSS. CpG-poor promoter sequences were defined as promoter sequences ranking at the bottom half based on their CpG density. Enhancer sequences were defined as sequences annotated in chromHMM as putative enhancers (i.e., enhancers, genic enhancers, and bivalent enhancer) in osteoblast cells.

In putative enhancer sequences we found a slightly weaker link between methylation and activity compared to promoter sequences, with 3% hypermethylation of downregulating sequences and 5% hypomethylation of upregulating sequences. Perhaps in accordance with the much weaker link between enhancer methylation and activity (*Jones, 2012*), this trend is not significant despite having similar statistical power to the promoter analysis (p=0.12, paired *t*-test). To test whether our results might have been affected by CpG density, we compared CpG density in differentially active compared to non-differentially active sequences, and in upregulating compared to downregulating sequences. We found no significant difference in CpG density between these groups (p>0.05, *t*-test).

The hypermethylation of downregulating sequences in modern compared to archaic humans and the hypomethylation of upregulating sequences in modern compared to archaic humans are also observed to some extent when testing these sequences in NPCs, but not in ESCs. For example, the top 10 upregulating sequences are hypomethylated by 7% on average in modern compared to archaic humans, top 10 downregulating sequences are hypermethylated by 13% in modern compared to archaic humans. This is in line with previous observations that differentially methylated regions tend to be shared across tissues (*Hernando-Herraez et al., 2013*).

## Differential TF binding sites

We predicted differences in binding of human TFs caused by each of our variants as follows. First, we downloaded the entire set of publicly available human TF binding motifs (7705 motifs, 6608 publicly available) from the Catalogue of Inferred Sequence Binding Preferences (CIS-BP) database (http://cisbp.ccbr.utoronto.ca/) and filtered them to include only motifs labeled as *directly determined* (i.e., we filtered out inferred motifs), resulting in 4351 motifs. Next, to enrich our mapping result for matches covering the variant location, we trimmed each of our oligo sequences containing a single variant to ±30 bp around the variant (the length of the longest motif). We did not trim oligos containing >1 variant. We used FIMO (*Grant et al., 2011*) to map each remaining motif to both the

archaic and modern alleles of each trimmed sequence (or untrimmed, for sequences with >1 variant). A background model was generated using fasta-get-markov using the trimmed (or untrimmed, if >1 variant) sequences. For each motif mapping to both the archaic and modern alleles at the same strand and location, we required that at least one allele had a q-value (as supplied by FIMO ≤0.05). Then, we found cases where the FIMO predicted binding score of a motif differed between the archaic and modern alleles. FIMO uses a p-value cutoff of $10^{-4}$ for reporting predicted binding. Therefore, some sequence pairs have a reported score for only one of the alleles. To assign these sequence pairs with a score difference, we used a conservative approach where we assigned the unscored allele with this lowest score reported for that motif, representing a score that is closest to a p-value of $10^{-4}$. Because the unreported score could be anywhere below the lowest reported score, but could not have been above it, this results in a conservative underestimation of the score difference. Finally, we linked each motif to the TF it is most confidently associated with in CIS-BP, thereby generating lists of TFs that showed differential predicted binding for each sequence. For cases in which multiple unique motifs corresponded to the same TF, we used the motif with the largest score difference between alleles. TF enrichment analyses were done on all predicted differential TF binding sites for TFs with a minimum of 10 predicted differential sites. TFs that are not expressed in the cell types we examined in this study (FPKM <1) were removed from the analyses. For TF expression in ESCs, we used ENCODE RNA-seq data for H1-hESC (*ENCODE Project Consortium, 2012*). For osteoblast expression, data (*Moriarity et al., 2015*) was downloaded from GEO under accession number: GSE57925. For NPC expression, data (*Lu et al., 2020*) was downloaded from GEO under accession number: GSE115407. Fisher's exact test was used to compute enrichment of a TF among differentially active sequences compared to other active sequences. p-Values were FDR-adjusted.

To further test the enrichment of ZNF281, we examined various cutoffs of the number of predicted bound motifs, ranging from 5 to a maximum of 14 (the number of motifs predicted to be differentially bound by ZNF281) in steps of 1. We found that with the exception of the cutoffs of 5 and 6 (where ZNF281 is only slightly above the significance threshold: FDR = 0.058 and 0.053, respectively), ZNF281 is the only significant TF across all of these cutoffs (FDR ≤0.05). We repeated the same test for FPKM cutoffs, ranging from 0.5 to 3 in steps of 0.5, and found that ZNF281 is the only significantly enriched TF (FDR ≤0.05) across all of these cutoffs.

For the predicted binding vs. expression correlation analysis, a cutoff of 10 sites per TF was used. p-Values were computed using Pearson's correlation.

## Overlapping loci with genomic features

The following datasets were used for the overlap analyses: GENCODE v28 GRCh38 human genome TSS (*Frankish et al., 2019*), GTEx v8 eQTLs (*GTEx Consortium, 2015*), and broad peaks for the following histone modification marks: H3K27ac, H3K4me1, H3K4me2, H3K4me3, H3K9ac, H3K9me3, and H3K27me, and the histone variant H2A.Z from the Roadmap Project for ESCs, ESC-derived NPCs, and osteoblasts (*Kundaje et al., 2015*). We overlapped each of these datasets with the lists of inactive and active sequences, and computed enrichment p-values using a Fisher's exact test. We repeated this for various RNA/DNA cutoffs (1, 1.5, 2, 2.5, 3, and 3.5). Sex chromosomes were removed from the analyses. p-values were FDR-adjusted using the Benjamini-Hochberg procedure.

Sequence conservation within primates was taken from the Altai Neanderthal genome annotation, which used the PhyloP metric (*Prüfer et al., 2014*).

## Human-chimpanzee *cis*-regulatory expression changes

We investigated the expression of genes associated with differentially active sequences by analyzing human and chimp RNA-seq data. As the expression changes we report are driven by *cis*-regulatory changes, we used our recently generated RNA-seq data from human-chimp hybrid cells (*Gokhman et al., 2021*) (GEO accession numbers: GSE146481 and GSE144825). In these hybrid cells, the human and chimpanzee chromosomes are found within the same nuclear environment and are exposed to the same trans factors (e.g., TFs). Therefore, any differential expression observed between the human and chimpanzee alleles within these hybrid cells is attributed to *cis*-regulatory changes. These cells are hybrid human-chimpanzee induced pluripotent stem cells (iPSCs), and we therefore investigated whether genes associated with upregulating sequences in our ESC lentiMPRA

data tend to be upregulated in the hybrid iPSCs and vice versa. It is important to note that differential expression between humans and chimpanzees reflects ~12 million years of evolution (i.e., changes that emerged along the human as well as along the chimpanzee lineages since their split from their common ancestor ~6 million years ago). However, our lentiMPRA data was done on sequences that changed along the modern human lineage (~550–765 thousand years). Therefore, the human-chimpanzee differences span an evolutionary time that is ~20-fold longer than the modern human lineage, and the effect of modern-derived variants on gene expression between humans and chimpanzees is expected to be largely diluted by the many other changes that accumulated along the rest of this time. Indeed, we observe a very slight, but significant correlation between differential expression observed in the lentiMPRA data and differential expression observed in the human-chimp hybrid data (p=0.017, Pearson's *r* = 0.1, *Figure 3—figure supplement 2g*).

## Phenotype enrichment analyses

Body part enrichment analyses were conducted using Gene ORGANizer v13. The analyses were conducted on sequences driving increased expression, sequences driving decreased expression, and all differentially active sequences. This was done in each of the three cell types. We conducted these analyses using various LFC thresholds: 0, 0.5, and 0.75, on the non-stringent locus-gene associations, and using a cutoff of five genes per term. Analyses were done against the active sequences as background and using the ORGANizer tool with the *confident* option. p-values were FDR-adjusted using the Benjamini-Hochberg procedure. For osteoblasts, non-skeletal organs were removed from the analyses. For NPCs, non-neuronal organs were removed.

For the HPO analyses, we used HPO (*Köhler et al., 2014*) build 1268 (November 8, 2019), analyzing gene lists identical to the Gene ORGANizer analyses, with the exception of using a cutoff of three genes per term, because fewer genes are linked to HPO terms than to Gene ORGANizer terms. Lists of phenotypes from HPO were generated for each variant through its linked genes. Hypergeometric test p-values were computed per phenotype and FDR-adjusted. Similarly to the Gene ORGANizer analysis, we removed non-skeletal phenotypes from the osteoblast results and non-neuronal phenotypes from the NPC results.

Gene Ontology, Gene ORGANizer, and HPO analyses were also done on the full set of genes linked to the 14,042 fixed variants using the same parameters described above (*Supplementary file 6*). Importantly, unlike the analyses of differentially active sequences, which can be compared against a non-differentially active sequences background to control for potential biases, the full set of sequences cannot be compared against a background set. Therefore, these results may be affected by different confounders such as GC content, the ability to call SNPs, DNA degradation patterns, and it is still to be determined to what extent these results reflect true evolutionary trends.

*SATB2* phenotypic analysis was done as previously described in *Gokhman et al., 2019*. In short, we used HPO (*Köhler et al., 2014*) build 1268 (November 8, 2019) to link phenotypes to *SATB2*. In addition, we conducted a literature search to expand gene-phenotype links to include studies that did not appear on HPO (*Supplementary file 5*). We used only skeletal directional phenotypes, that is, phenotypes that could be described on a scale (e.g., smaller/larger hands), as these could be examined against the fossil record. This resulted in 34 phenotypes that are the result of *SATB2* heterozygous LOF (*Supplementary file 5*). Phenotypes that are included in another phenotype (e.g., *prominent nasal bridge* and *prominent nose*) were merged, and contradicting phenotypes (e.g., *broad nose* and *thin/small nose*) were removed. This resulted in a final list of 17 phenotypes (*Supplementary file 5*). Given that the mechanism underlying these phenotypes is a decrease in the dosage of SATB2, and that *SATB2* is possibly downregulated in modern humans, we sought to investigate if similar phenotypes exist between modern human patients with *SATB2* heterozygous LOF and archaic humans. For each phenotype, we determined if it is divergent between the modern and archaic humans based on previously published annotation (*Gokhman et al., 2019*). Then, for remaining divergent phenotypes, we tested if the direction between patients and healthy individuals matches the direction between modern and archaic humans. The significance of directionality match was computed using a binomial test, with a random probability of success p=0.5. To compute the significance of the overall number of phenotypes that are divergent and match in direction, we compared the overall number of annotated divergent phenotypes to the number of divergent phenotypes associated with *SATB2* using a hypergeometric test. Out of a total of 696 annotated

phenotypes between modern and archaic humans (*Gokhman et al., 2019*), 434 are annotated as divergent, and the direction of 50% of them (217 phenotypes) is expected to match by chance.

## Acknowledgements

We would like to thank Tal Ashuach (MPRAnalyze), Terence Capellini, Evelyn Jagoda, Martin Kircher, and the Fraser, Petrov, and McCoy labs for helpful feedback. DG was supported by the Human Frontier, Rothschild, and Zuckerman fellowships. This work was supported in part by the National Human Genome Research Institute grant 1UM1HG009408 (NA), the National Institute of Mental Health grants 1R01MH109907 (NA) and 1U01MH116438 (NA), the Uehara Memorial Foundation (FI), and the Stanford Center for Computational, Evolutionary and Human Genomics (CEHG).

## Additional information

### Funding

| Funder | Grant reference number | Author |
| --- | --- | --- |
| National Human Genome Research Institute | 1UM1HG009408 | Nadav Ahituv |
| National Institute of Mental Health | 1R01MH109907 | Nadav Ahituv |
| National Institute of Mental Health | 1U01MH116438 | Nadav Ahituv |
| Uehara Memorial Foundation | | Fumitaka Inoue |
| Stanford Center for Computational, Evolutionary and Human Genomics | | Carly V Weiss |

The funders had no role in study design, data collection and interpretation, or the decision to submit the work for publication.

### Author contributions

Carly V Weiss, Data curation, Formal analysis, Validation, Investigation, Visualization, Methodology, Writing - original draft, Writing - review and editing; Lana Harshman, Resources, Data curation, Formal analysis, Validation, Investigation, Visualization, Methodology, Writing - original draft, Writing - review and editing; Fumitaka Inoue, Data curation, Validation, Investigation, Methodology, Writing - original draft, Writing - review and editing; Hunter B Fraser, Supervision, Writing - original draft, Writing - review and editing; Dmitri A Petrov, Supervision, Funding acquisition, Writing - original draft, Writing - review and editing; Nadav Ahituv, Conceptualization, Resources, Data curation, Supervision, Funding acquisition, Validation, Investigation, Methodology, Writing - original draft, Writing - review and editing; David Gokhman, Conceptualization, Data curation, Software, Formal analysis, Supervision, Validation, Investigation, Visualization, Methodology, Writing - original draft, Writing - review and editing

### Author ORCIDs

Fumitaka Inoue (iD) http://orcid.org/0000-0003-0657-434X
Hunter B Fraser (iD) http://orcid.org/0000-0001-8400-8541
Dmitri A Petrov (iD) http://orcid.org/0000-0002-3664-9130
Nadav Ahituv (iD) http://orcid.org/0000-0002-7434-8144
David Gokhman (iD) https://orcid.org/0000-0002-3536-9006

### Decision letter and Author response

Decision letter https://doi.org/10.7554/eLife.63713.sa1
Author response https://doi.org/10.7554/eLife.63713.sa2

# Additional files

## Supplementary files

• Supplementary file 1. Modern-derived fixed sequences and their massively parallel reporter assay (MPRA) expression data, linked genes, and chromatin annotations.

• Supplementary file 2. Overlap of active sequences with various genomic features indicative of transcriptional activity.

• Supplementary file 3. Predicted transcription factor binding alterations and their correlation with expression.

• Supplementary file 4. Enriched Human Phenotype Ontology (HPO), Gene Ontology, and Gene ORGANizer terms within differentially active sequences.

• Supplementary file 5. Human phenotypes associated with SATB2 heterozygous loss-of-function and their respective state in modern humans vs. Neanderthals.

• Supplementary file 6. Gene Ontology, Human Phenotype Ontology (HPO), and Gene ORGANizer terms enriched within genes associated with modern human-derived fixed variants.

• Transparent reporting form

## Data availability

Data was deposited in GEO under accession number: GSE152404.

The following dataset was generated:

| Author(s) | Year | Dataset title | Dataset URL | Database and Identifier |
|---|---|---|---|---|
| Weiss CV, Harshman L, Inoue F, Fraser HB, Ahituv N, Gokhman D | 2020 | MPRA | https://www.ncbi.nlm.nih.gov/geo/query/acc.cgi?acc=GSE152404 | NCBI Gene Expression Omnibus, GSE152404 |

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
