## [Decision Letter]

**Acceptance summary:**

This study advances our understanding of human evolution by testing the function of human-specific variants in non-coding regions that regulate gene expression. In some cases, variants are identified that might contribute to traits limited to modern humans. The comprehensiveness of dataset should also make this work an important community resource.

**Decision letter after peer review:**

Thank you for submitting your article "The cis-regulatory effects of modern human-specific variants" for consideration by *eLife*. Your article has been reviewed by 3 peer reviewers, and the evaluation has been overseen by Patricia Wittkopp as the Senior and Reviewing Editor. The reviewers have opted to remain anonymous.

The reviewers have discussed the reviews with one another and the Reviewing Editor has drafted this decision to help you prepare a revised submission.

All three reviewers and I found much to like in this work. In the post-review discussion, there was general agreement of the strengths and weaknesses in the study, with the most pressing issue being the need to be more clear about the caveats resulting from assaying fragments with individual variants, out of their native genomic context, and in a limited number of cell types. These must be addressed directly in a revised version. Beyond this, each reviewer also identified additional points that need clarification and refinement. Because of the depth of these comments, I am including all three reviewer's comments below in their entirety. I look forward to seeing a revised version of this work.

Reviewer #1:

Weiss and colleagues performed a lentiMPRA comparing archaic and human genetic variants in three cell types. I found this study well designed and executed and the results exciting. I especially liked the analysis of SATB2! It is really amazing that by focusing on skeletal phenotypes the authors can root inferences from their study in human biology. I wish this was done for more than one gene! I still have a number of concerns about the study, outlined below.

1. In a 200 bp sequence background, there are typically other variants included. If I understand correctly, the authors only considered one focal variant that differs between modern and archaic alleles and tested for a difference in one of the two possible sequence backgrounds. This means that one variant is tested in an artificial allele that possibly never existed in an archaic or modern human. in vivo, this variant might interact with nearby variants (fixed or segregating), but this is ignored in this setting. It would obviously be too much to demand that both the modern and the archaic variant be tested in the archaic allele as well, but I wonder if the authors can report if there are cases (and if so, how many) where tested variants are close to other variants which would indicate that the result is artificial and thus unreliable. The relevant distance would be within the size of one transcription factor binding site or less than 15-20 bp or so.

2. Line 179/Figure 2, There is a larger overlap between NPCs and osteoblasts than between either and ESCs – due to either cell type's closer developmental similarity with ESCs, I had expected the opposite. Can the authors offer an explanation?

3. Line 176, How often were both archaic and modern alleles active and how often only one?

4. Line 192, "At the same time, these sequences tended to be depleted of repressive marks". Figure 2b shows an enrichment of H3K27me3 in active sequences, while no depletion for H3K9me3. Figure 2c shows a mild depletion at best and Figure 2d a combination of enrichment and depletion of both marks. It seems to me that the quoted statement is not supported by the data.

5. Figure 3, There are a few differentially active sequences that show very low fold changes. I recommend using a cutoff to remove sequences with such small fold changes. What would be the biological significance of a sequence that is barely different between alleles but considered significantly differentially active? Related, if there are sequences that show such small fold changes, it is possible that there are some differentially active sequences that disagree in the direction of bias between replicates. If there are any, those should certainly be excluded!

6. Line 504, As most chromatin regions are inactive in most cell types, the use of insulators may more often than not lead to an increase in activity and thus to false positive detection of regulatory activity in lentiMPRA. This means, importantly, that for many of the active sequences in this study, there is no evidence that they would be active in this cell type at all, because not all sequences overlap a signature of activity (histone mark, active chromatin etc.). Those sequences might only be active because of the use of insulators. I think all that can really be said of those sequences is that they have the potential to actively drive transcription in a hypothetical cell type in which they would be accessible and active.

7. Lines 251-252, "[…] suggests that these sequences are likely to be differentially active in other cell types not assayed in this lentiMPRA." I disagree with this idea. I would argue that this indicates that regulatory logic is independent of cell type but mainly determined by the underlying sequence which determines which transcription factors can bind. That sequence will bind the same or related transcription factors expressed in two different cell types which will lead to similar effects in distinct cell types. But this would also be true for a sequence that is only active in cell type A but heterologously expressed in cell type Y. Especially due to the use of the insulator elements in the construct, we don't know if the sequence would really be active.

8. Line 317, It appears that this analysis does not differentiate between activating and repressive transcription factors. I would expect that repressive transcription factors invert this relationship. Do the authors find that to be true?

9. Lines 332-336, The language here indicates that the authors looked for positive correlations only ("i.e., higher affinity to the modern human sequences was predictive of higher expression"). Then they state that "All of these TFs had a positive correlation" which is trivial if they only looked for positive correlations. It should be clarified if negative correlations were investigated as well.

10. Line 342, The finding that ZNF281 and SP3 are significantly enriched among differentially active sequences in NPCs at an FDR of 0.05 is only corrected for the tests done in NPCs. However, the authors tested for enrichment among all three cell types (plus the union of all three cell types together). This means that multiple testing correction should have been performed over the entire dataset that was used in the test and not separately by cell type. If done over the entire dataset (all three cell types, ignoring the union of all cell types), the FDR for ZNF281 is 0.056 and that of SP3 is 0.13. This means that there is no enrichment of any transcription factors among differentially active sequences after appropriate multiple testing correction. I did not check this for the other analyses, but multiple testing correction should be performed over the entire analyzed dataset and not per cell type throughout the study, for instance for enrichment of GO, HPO etc.

11. Line 551-557, This seems like a gross overinterpretation to me. If a binding site is detected or not given one base pair, is most closely related to the significance cutoff for detection of a binding site in the software being used. This could also be interpreted such that if it depends on one single base pair if there is binding or not, there might not be much binding at all. That the cell regulatory machinery uses the same cutoff for binding as the method to detect binding in this study is extremely unlikely. Further, binding is not only determined by the sequence itself but also by sequence activity and accessibility. In other words, the evolution of a new putative binding site in inaccessible heterochromatin is likely inconsequential. It would be more likely for a newly evolved, putative binding site to be active adjacent to an already existing, active enhancer, where the sequence would already be accessible. But lentiMPRA is not appropriate for detecting such events and speculations about newly evolved regulatory elements seem unwarranted.

12. Line 621-622, How many of the variants included in the study have information in only one of the archaic human genomes?

13. Lines 636-639, The way how the design of the oligos is described is very confusing and maybe actually misleading. For instance here: "For each variant we designed two fragments, one with the ancestral sequence and one with the derived sequence." I don't think that is correct. If I understand correctly, it should say "For each variant we designed two fragments, one with the ancestral variant state and one with the derived variant state, each in the background of the modern human [Is this correct?] sequence." because sequence versions do not differ in their entire sequence but only in the single variant whereas the sequences are otherwise identical.

14. Line 905, Where TFs filtered for expression in the analyzed cell type? If not, that should be done!

15. Line 906-908, "For cases in which multiple unique motifs corresponded to the same TF, we used the motif with the largest score difference between alleles." I don't see why that would make sense. If there are different score differences for different motifs on the same allele, wouldn't the expectation be that in each case the highest score binds? That could mean that a substitution would swap one motif for another, such that the binding differential of either motif is irrelevant to the question but what really matters instead is the binding differential between the respective maximum binding affinity of the two motifs.

Reviewer #2:

Weiss et al. screened 14,042 variants that differ between modern and archaic humans genome-wide for their gene-regulatory impact. Using massively parallel reporter assay (lentiMPRA) in three cell types, they find evidence that ~13% of genomic sequences harboring variants can drive gene expression (=active sequences), and that about 13% * 23% ≈ 3% of variants lead to differential regulatory activity (=differentially active sequences). The authors study, characterize and compare active and differentially active sequences with regard to genome annotations; they find, for example, that differentially active sequences are enriched in annotated enhancer loci. With regards to a possible mechanism for differential regulatory activity observed, Weiss et al. study differences in transcription factor (TF) binding site motifs between corresponding archaic vs. modern sequences, and they are able to report substantial motif changes for ZNF81 and SP3. Linking first variants to genes and then second genes to organs and phenotypes allows the authors to relate putative variants' effects to brain and vocal tract; then a variant with reduced regulatory activity in the modern human version (across all there cell types studied) is discussed in more detail, with the conclusion that reduced SATB2 expression could give rise to brain and craniofacial phenotypes different between modern and archaic humans.

Effects of (modern) human-specific variants are of great interest, so this study using powerful MPRA technology for a comprehensive assessment of all variants is exciting and a step forward. The authors are able to suggest specific variants that may underlie modern human traits, and overall the manuscript is clear. The linking of variants to organs and phenotypes is interesting, and it provides several potential entry points for follow-up studies. Importantly, the data provided constitute a significant resource for the community and will have an impact on further studies in this area.

While it would be great to have a more definite linking of variants to genes, in general and specifically in the case of SABT2, where there is no direct experimental evidence that chr2:199,469,203 affects that gene's expression, that would require additional experiments. An easier improvement would be for the authors to publish the computer code used to generate results; currently this is not the case, nor do the authors state that it is made available upon (reasonable) request. This makes it very hard to reproduce results reported, study details of the analyses performed, or explore effects of parameter choices, thereby diminishing the value/impact of the study. The manuscript would benefit from addressing the following specific points.

1. P8 / Supplementary Figure 1: Panel B: By eye the data looks quite a bit more noisy for ESC than for the other two cell types. Correlation is less for ESC, and seems driven by only few sequences. Additionally reporting Spearman correlation would be good, and I this should be discussed in the manuscript. Panel D: It would be more informative using a log-scale for the RNA/DNA ratio. Also, it would be informative to see the tested sequences together with the scrambled and the positive.

2. P9 L 169: It might be interesting for the authors to provide an overview of where/of what type the sequences are. For example., are they mostly intergenic, proximal, or inside genes? If so, is there perhaps an enrichment for certain types of genes, etc.? Even in case not novel, where archaic vs. modern human differences are would be good to know in light of contrasting active vs. inactive sequences later in the manuscript.

3. P9 L 172 / Methods P38-40 L761-802: While the authors describe their MPRA analysis, safe for actually attempting to re-do the analysis, it remains unclear whether sufficient details are provided to reproduce results; publishing code used for the analysis would be far better. Further on, processed data provided on GEO has replicates already aggregated, so that it cannot be used to reproduce, e.g., the MPRAnalyze part of the analysis and primary data would have to be re-processed first. Also, it is unclear why the authors use a combination of MPRAnalyze and a heuristic involving aggregation across replicates to assign oligos/sequences as "active". It'd be good to explain why that is done, and what is the effect on the set of active oligos.

4. P10 L 190-192: Figure 2: Panels B-D: For each bin of RNA/DNA ratio enrichment appears to be plotted, based on a hypergeometic/Fisher test, probably as the logarithm of the odds ratio. It would be informative to plot a measure of variability/confidence for each bin as well. Standard implementation of the Fisher test in R provides a confidence interval, for example.

5. P10 L 198-200, Figure 2: Similar to the point above, it would be good to see "error bars" on panel e.

6. P12 L 235: It is unclear what the null model for the Super Exact test is. Is overlap of differentially active sequences between cell types higher than expected, given what we already know about the overlap considering all active sequences? Or do cell types overlap for differentially active sequences more than expected given sequences were selected purely at random, but not as much as they do considering all active sequences?

7. P15 L 272-274: Supplementary Figure 3: Variability between lentiMPRA replicates is not displayed in panel B; it would be good to use a display that shows variability in replicates of both arrays, lentiMPRA and luciferase assay.

8. P16 L 300-305: For the methylation analysis, it there a cell type signal in the sense that up-regulation correlates with hypomethylation and down regulation with hypermethylation in osteoblasts, but not in the other two cell types?

9. P17 L 322-325: Supplementary Figure 4: What exactly is plotted in panels B-D: does each point correspond to a sequence-TF combination, or was some kind of averaging performed (over sequences or TFs)?

10. P44 L 900-903: FIMO analysis: Unclear why the lowest-matching score would be used. Why not use the highest log odds on that allele? Also, what background model was used for FIMO?

11. P18, L 342: How sensitive are the results of the enrichment analysis (ZNF281 and SP3) with regards to some of the parameters, like requiring that 10 or more motif changes are required for TFs to be considered? How many TFs did meet that requirement?

12. P20 L 371-376: In the supplementary table, it would be good to provide information about how a gene was linked to a variant (i.e., which of the four lines of evidence used support the link). Also, for the Hi-C data, the authors describe two lists in the methods a cell type-specific list and a general list. However, it appears unclear which one was used for linking to genes.

13. P20 L 387-391 and P21 L 394-396: Which genes are driving theses terms? Do they largely overlap between different terms, or are they different genes in each term. It would be informative to report genes alongside terms in the supplemental table. Also for the HPO analysis in the following paragraph.

14. P23 L 432: PyloP conservation scores are usually -log10(p-value), so while conservation is supported, the p-value corresponding to a score of 0.996 would be about 0.1, which is not that highly conserved.

15. P25 L 484-486. The authors write they have investigated how single nucleotide variants "have affected expression". While technically true in the context of the lentiMPRA, the authors measure changes in the ability of these sequences to drive expression, but do not show that they actually affect "real live" gene expression outside of an artificial construct. Perhaps change this sentence.

Reviewer #3:

The majority of sequence variants that separate modern and archaic human genomes are found in non-coding regions, often overlapping gene regulatory elements. It is difficult to study the functional significance of these variants on molecular phenotypes such as transcription and chromatin regulation due to the lack of intact biological material in archaic specimens. The current work uses an MPRA framework to compare the gene regulatory activities of derived modern human versus ancestral variants in three modern human cell types – ESCs, NPCs and osteoblasts. As expected, the authors found that active sequences were associated with enhancer annotations, and differentially active sequences were associated with divergent TF binding motifs. The presented MPRA data is well controlled. However, the differences detected by the various comparisons made were often subtle, or small in magnitude. This requires robust statistical analysis to understand the significance of those differences. While the authors were moderate in most of their claims, this reviewer thinks some of those claims require further analytical support with some adjustments in the text. With these additional non-experimental, but critical analytical revisions and some clarifications, this would be of potentially high interest to a broad readership as a contribution to *eLife*.

1. Selection of cell types – this is crucial as TFs are the major drivers of MPRA activity. The claim that ESCs were used "due to their globally active transcription, providing a general view of gene expression" is based on a single study from mouse embryonic stem cells (Efroni et al., 2008). Mouse ESCs are known to have very different biological properties from human ESCs, with mouse ESCs representing a less differentiated state than human ESCs (Hanna et al., PNAS 2010). Therefore, this is not a sufficient justification for using ESCs as a comparative cell type. In particular, enhancers are typically cell type or lineage specific in their activity, and they are driven by TFs that are expressed in a lineage dependent manner. hESCs express basal and pluripotency factors, limiting their capacity to drive transcription of developmentally specialized enhancers. The use of NPCs and osteoblasts are reasonable considering the central interest of the study in understanding the role of derived alleles in shaping modern human phenotypic differences, but beyond that point it is not clear why the cell types were chosen (see point 2).

2. Of the 14000 regions selected for the MPRA – what was the expectation given existing functional genomic data from modern humans and how did that inform the selection of cell types? This question is related to the above point. In supplementary figure 2 the authors provide the annotation classifications of the 14,000 fixed derived variants based on ChromHMM. This data indicates that ~25% of these variants overlap enhancers or TSS regions – but the relative distributions of the annotations differ depending on the ChromHMM context (ESC, NPC or Osteoblasts). It seems that using this methodology, one can predict other cell-types for which these variants may be active, thus providing an informed context to test the MPRA. It is important to know what percentage of the fixed variants are located in any type of regulatory region across a variety of cell-types. This information may also be important for understanding differential activity of modern vs archaic sequences, as the magnitude of differential activity may be cell type dependent. Related to this point, in the Discussion the authors point out that, based on previous studies, trans differences are likely minimal between modern and archaic humans, but what about trans differences in cell-type environment? Their assertion is perhaps an overgeneralization that diminishes the importance of trans effects. This is a point that should be addressed.

3. The authors report a modest median fold-change of 1.2x for differentially active sequences. They use a combination of criteria to associate these sequences with a putative target gene. An important next step to support the significance of these findings would be to understand whether these modest differences are linked to differential expression of target genes. Obviously, such a comparison cannot be made for modern and archaic human transcriptomes; however, one potential avenue for testing this idea would be to use public data from comparative human and chimp RNA-seq studies to look at whether the target genes are differentially expressed between modern human and chimp, provided that the ancestral allele for the differentially active sequence is found in chimp.

4. The section describing the relationship between DNA methylation and differentially active sequences seems to be one of the least well supported pieces of this study. In the absence of gene expression data, a key piece of support for the functional significance of differentially active sequences is to link those differences to epigenetic differences. The availability of DNA methylation maps from Neanderthal and Denisovan makes this possible. However, the way this data is presented (or not presented) is really difficult to interpret (i.e. reporting that upregulating sequences tend to be hypomethylated with a -1% difference compared to what?). A figure would have been very useful here. Also, it's important to note that many enhancers have low CpG density, while promoters and TSSs with high CpG density are typically hypomethylated regardless of gene activity (Lister et al. 2009, Stadler et al. 2011, Schlesinger et al. 2013). Thus, it is necessary to parse these regions when referring to methylation differences.

---

## [Author Response]

Reviewer #1:Weiss and colleagues performed a lentiMPRA comparing archaic and human genetic variants in three cell types. I found this study well designed and executed and the results exciting. I especially liked the analysis of SATB2! It is really amazing that by focusing on skeletal phenotypes the authors can root inferences from their study in human biology. I wish this was done for more than one gene! I still have a number of concerns about the study, outlined below.

We thank the reviewer for their feedback and are glad that they found this work exciting. We share the reviewer’s excitement for the prospect of identifying additional candidate genes, and we trust that this data will serve as an important resource for the identification of genes underlying modern human evolution.

1. In a 200 bp sequence background, there are typically other variants included. If I understand correctly, the authors only considered one focal variant that differs between modern and archaic alleles and tested for a difference in one of the two possible sequence backgrounds. This means that one variant is tested in an artificial allele that possibly never existed in an archaic or modern human. in vivo, this variant might interact with nearby variants (fixed or segregating), but this is ignored in this setting. It would obviously be too much to demand that both the modern and the archaic variant be tested in the archaic allele as well, but I wonder if the authors can report if there are cases (and if so, how many) where tested variants are close to other variants which would indicate that the result is artificial and thus unreliable. The relevant distance would be within the size of one transcription factor binding site or less than 15-20 bp or so.

Following the reviewer’s comment, we realized that this point required further clarification. Most importantly, each sequence included either the ancestral sequence throughout, or the derived sequence throughout. In other words, if a 200 bp window had more than one variant separating modern from archaic humans and apes, the other derived variants were also included in the derived construct. Thus, each focal modern-derived variant was tested against a modern human background, and each focal ancestral variant was tested against an ancestral background in the same experiment. Notably, only 1,362 of the 14,042 pairs of sequences (10%) had >1 fixed variant. Following the reviewer’s comment, we realized that we did not adequately describe this design in the text, and we therefore added the following section to the main text: “13,680 out of 14,042 sequence pairs (90%) had a single variant separating the human groups. For the 1,362 sequence pairs containing additional variants within the 200 bp window, we used either the modern-only or archaic-only variants throughout the sequence”.

As the reviewer mentioned, in addition to the fixed variants we investigate, these sequences may contain additional non-fixed variants in some modern or archaic individuals. We agree with the reviewer that these non-fixed variants may have an effect on the activity of some of these sequences. Because in these cases both alleles of the non-fixed variants exist in the modern human population, these constructs reflect sequences that likely exist in at least some humans. To emphasize these caveats, we added the following section to the discussion of the MPRA limitations: “Sixth, the level of sequence activity may depend on more than one variant (including non-fixed variants, which we have not tested here). In the cases of non-fixed variants, the extent of differential activity could vary between individuals. At the same time, in the 10% of sequences that include more than one fixed variant, in our study it is not possible to determine which of the variants drives the differential activity (with the exception of cases with more than two variants where the tiled sequences include a different combination of these variants).”

2. Line 179/Figure 2, There is a larger overlap between NPCs and osteoblasts than between either and ESCs – due to either cell type's closer developmental similarity with ESCs, I had expected the opposite. Can the authors offer an explanation?

Like the reviewer, we were also expecting more overlap between each cell type and ESCs than to each other. Reassuringly, when looking at the differentially active sequences (Figure 3d) compared to all active sequences (Figure 2a) this trend is far less extreme. Regardless, we believe technical reasons could partially explain the observed trend. ESCs are very sensitive to infection (see methods where we note the MOI used for ESCs was 10 compared to 50 for the other cell types and that “We used a lower MOI for ESC because the cells are very sensitive to infection and a MOI higher than 10 would result in cell death”; L807). Due to the difficulty in infection, the quality of the data was slightly lower for ESCs than for the other cell types. To this point, we added the following to the Results section: “We saw a strong correlation of RNA/DNA ratios between replicates for all cell types (Pearson’s r = 0.76 – 0.96, *P* < 10^-100^ , Supplementary Figure 2b), with the lower correlation scores being in ESC, likely due to our use of lower multiplicity of infection (MOI) in these cells due to their increased sensitivity to lentivirus infection”. We also note in the methods that we required additional sequencing for ESCs “due to lower lentivirus infection efficiency in this cell type.”. We believe the technical difficulties with these cells could be a reason we see less of an overlap between ESCs and other cell types. Importantly, we make a point in the discussion to clarify that “it is difficult to determine what proportion of [variation] is due to biological versus technical factors” and “Thus, we largely refrained from comparisons between cell types”.

Additionally, the reason that both we and the reviewers expected this specific trend rests on the assumption that the test sequences are a representative sampling of functional sequences across the genome and therefore can be considered representative of activity in a given cell type. Although our target sequences were selected in an unbiased manner, there may be, by chance, some unknown bias, especially given the low number of active sequences. This bias could be the cause for an unexpected overlap in activity. To clarify this point, we’ve revised the previously mentioned sentence to read “Thus, we largely refrained from comparisons between cell types and the overlap observed in Figure 2a and Figure 3a should not be used to define such similarities. Rather, these diagrams should be used to examine the replicability of our results.”

3. Line 176, How often were both archaic and modern alleles active and how often only one?

In ESCs: 439 sequences were significantly active only in the archaic allele, 447 only in the modern allele, and 297 in both. In osteoblasts: 210 only in the archaic allele, 178 only in the modern allele, and 426 in both. In NPCs: 126 only in the archaic allele, 178 only in the modern allele, and 426 in both. Importantly, the absence of evidence of activity of the other allele is likely not evidence of inactivity, but rather could be the result of marginally significant activity or limited power to detect activity: the mean FDR for alleles that were identified as active is 0.01 (ESCs and osteoblasts) and 0.02 (NPCs). The full data appears in Supplementary File 1. We now refer to this supplementary table in the paragraph introducing our analysis of active sequences: “We found that in ESCs, 8% (1,183) of sequence pairs drove expression in at least one of the alleles, 6% (814) in osteoblasts, and 4% (602) in NPCs (FDR < 0.05, Supplementary File 1, see Methods)”

4. Line 192, "At the same time, these sequences tended to be depleted of repressive marks". Figure 2b shows an enrichment of H3K27me3 in active sequences, while no depletion for H3K9me3. Figure 2c shows a mild depletion at best and Figure 2d a combination of enrichment and depletion of both marks. It seems to me that the quoted statement is not supported by the data.

H3K27me3 indeed shows an enrichment in ESCs. This is likely due to the fact that together with H3K4me3, the H3K27me3 modification in ESCs marks bivalent genes, which become active in later stages of differentiation (Bernstein B.E. et al., Cell, 2006, and Blanco et al., Trends in Genetics, 2020). To clarify the role of this modification in ESCs, we added the following statement to the figure legend: “The enrichment of H3K27me3 in ESCs possibly reflects the presence of this mark in bivalent genes, which become active in later stages of development (Blanco et al., Trends in Genetics, 2020).”

While H3K9me3 may not appear to be depleted when looking at the graph, our analysis shows a slight depletion that is significant (FDR = 0.02 for a minimum RNA/DNA ratio = 1). As the reviewer mentioned, in Figure 2c the depletion is mild, but here too – it is often significant (e.g., FDR = 0.006 for a minimum RNA/DNA ratio = 1 for H3K9me3) and becomes more substantial as the RNA/DNA cutoff increases. In Figure 2d the marks are indeed enriched in lower cutoffs but become depleted in higher cutoffs. We realize that our previous statement did not reflect the complex dynamics observed in these plots. Therefore, we rephrased it to “these sequences tended to show relatively fewer repressive marks compared to active marks (Figure 2b-d, Supplementary File 2)”. We have also added *p*-values and FDRs to this supplementary table.

5. Figure 3, There are a few differentially active sequences that show very low fold changes. I recommend using a cutoff to remove sequences with such small fold changes. What would be the biological significance of a sequence that is barely different between alleles but considered significantly differentially active?

We agree with the reviewer that some of these fold-changes might represent biologically insignificant changes. At the same time, any cutoff used here (in addition to the statistical cutoff we use) or in any other MPRA will likely be somewhat arbitrary, and for some sequences, even small fold changes might have a biological effect. As also shown both in these lentiMPRAs and other published lentiMPRA datasets (see for example, Inoue et al. Genome Research 2017, Kircher et al. Nature Communications 2019, Inoue et al. Cell Stem Cell, Klein et al. Nature Methods 2020), while the fold-changes can be small, in particular for enhancers, they are consistent across replicates. Also, of note, is that an advantage of our use of lentiMPRAs in this study is that we can quantitatively compare the modern and archaic human sequence side-by-side in the same lentiMPRA experiment to identify sequence changes that change their activity. Additionally, it is possible that for some genes, the effect per sequence is small, but the effects of several sequences might add up to a substantial effect on the gene. To clarify this, we added the following statement to the description of the limitations of MPRA: “At the same time, some minimally active sequences may not be biologically significant”.

Related, if there are sequences that show such small fold changes, it is possible that there are some differentially active sequences that disagree in the direction of bias between replicates. If there are any, those should certainly be excluded!

MPRAnalyze, which we used to determine the significance of differential activity, takes into account in its statistical analyses the level of agreement between replicates. Additionally, when comparing the agreement between different cell types (which is expected to be lower than between replicates), we observed a strong agreement, with 107 out of 109 sequences going in the same direction (*P* = 9.2x10^-30^, Binomial test). We thus rephrased this section: “We identified 109 sequence pairs that were differentially active in more than one cell type. Out of these 109, we found that 107 show the same direction of differential activity across cell types (*P* = 9.2x10^-30^, Binomial test), and we also observed a high correlation between the magnitudes of differential activity (Pearson’s *r* = 0.82, *P* = 1.6x10^-27^).”

6. Line 504, As most chromatin regions are inactive in most cell types, the use of insulators may more often than not lead to an increase in activity and thus to false positive detection of regulatory activity in lentiMPRA. This means, importantly, that for many of the active sequences in this study, there is no evidence that they would be active in this cell type at all, because not all sequences overlap a signature of activity (histone mark, active chromatin etc.). Those sequences might only be active because of the use of insulators. I think all that can really be said of those sequences is that they have the potential to actively drive transcription in a hypothetical cell type in which they would be accessible and active.

We agree with the reviewer that some of the sequences that show activity in our assay may in fact be inactive in their endogenous genomic context. Importantly, the majority of active sequences (86%) show at least one active histone modification peak. This is of course insufficient to determine whether these sequences are active in their endogenous genomic context. At the same time, the lack of active marks in some sequences does not necessarily mean that they are inactive. We refer to this point in the discussion: “while others might show activity that they otherwise would not have”. Following the reviewer’s comment, we added the following sentence to the “Characterization of active regulatory sequences” section: “Some of these sequences may show activity in the lentiMPRA experiment, but not in their endogenous genomic context”.

7. Lines 251-252, "[…] suggests that these sequences are likely to be differentially active in other cell types not assayed in this lentiMPRA." I disagree with this idea. I would argue that this indicates that regulatory logic is independent of cell type but mainly determined by the underlying sequence which determines which transcription factors can bind. That sequence will bind the same or related transcription factors expressed in two different cell types which will lead to similar effects in distinct cell types. But this would also be true for a sequence that is only active in cell type A but heterologously expressed in cell type Y. Especially due to the use of the insulator elements in the construct, we don't know if the sequence would really be active.

We agree that the observation that a sequence is differentially active in one cell type does not necessarily mean that this sequence is also differentially active, or even just active, in other cell types. Our claim referred to the observation that the magnitude of differential activity was significantly correlated between cell types (Pearson’s R = 0.82, *P* = 1.6x10^-27^), and that sequences tended to be differentially active in more than one cell type (overlap of 75-fold higher than expected, *P* < 10^-100^, Super Exact test, Figure 2a). In other words, it is unlikely that these sequences are differentially active only in the cell types we examined. To further test this, we analyzed eQTL data from GTEx v8. We tested how often eQTLs detected in the brain cortex are significant only in this tissue. We found that out of 500,000 randomly chosen variant-gene associations, only 113 (0.02%) were significant solely in the brain cortex. We repeated this for cultured fibroblasts and liver and found similar results (<1% of fibroblast and liver eQTLs were significant only in that cell type/tissue). Therefore, despite the complex and cell-type specific interactions between sequence and transcription factors, these interactions are often shared by more than one cell type.

8. Line 317, It appears that this analysis does not differentiate between activating and repressive transcription factors. I would expect that repressive transcription factors invert this relationship. Do the authors find that to be true?

We agree with the reviewer that increased binding by a repressor TF is expected to decrease, rather than increase, the activity of the bound sequence. Although we did not limit the analysis to activator TFs, the inherent design of MPRA is less likely to detect transcriptional repression due to the low basal activity of the minimal promoter used. Therefore, sequences that get bound by repressors and decrease the activity of the construct are less likely to be identified as active or differentially active (Tewhey et al., 2016). Indeed, all of the significant correlations we found between differential binding and differential activity were positive (Supplementary File 3). We also investigated previous studies of the TFs that came up in our analyses and found that none of them are exclusively a repressor, and most are predominantly or exclusively activators (with the exception of ZNF880, for which we did not find evidence establishing its role in transcriptional regulation). For examples of studies establishing the transcriptional role of these TFs, see the following PMIDs:

SP1 – 27697431

ZNF341 – 29907690

VEZF1 – 23921720

ZNF263 – 19887448

MAZ – 12684688

SP3 – 27697431

ZBTB17 – 29616049

ZNF281 – 32512343

EGR2 – 27856665

KLF6 – 32998281

All of which are now cited in the text. We now also emphasize in this section the limitation of the MPRA design in finding transcriptional repression: “…the use of a minimal promoter with basal activity in the MPRA design means that transcriptional repression is less likely to be detected, and therefore, further investigation is required in order to identify potential repressive activity in these sequences (see Discussion).”

9. Lines 332-336, The language here indicates that the authors looked for positive correlations only ("i.e., higher affinity to the modern human sequences was predictive of higher expression"). Then they state that "All of these TFs had a positive correlation" which is trivial if they only looked for positive correlations. It should be clarified if negative correlations were investigated as well.

Following the reviewer’s comment, we realized the text was unclear. We tested both positive and negative correlations in this analysis. To clarify this, we rephrased this section to: “we examined the correlation between binding and expression fold-change (either positive or negative). We found that changes to the motifs of 14 TFs were predictive of expression changes (Supplementary Figure 5d, Supplementary File 3). All of these TFs had a positive correlation between changes in their predicted binding affinity and changes in expression of their bound sequences, reflective of their known capability to promote transcription.”.

10. Line 342, The finding that ZNF281 and SP3 are significantly enriched among differentially active sequences in NPCs at an FDR of 0.05 is only corrected for the tests done in NPCs. However, the authors tested for enrichment among all three cell types (plus the union of all three cell types together). This means that multiple testing correction should have been performed over the entire dataset that was used in the test and not separately by cell type. If done over the entire dataset (all three cell types, ignoring the union of all cell types), the FDR for ZNF281 is 0.056 and that of SP3 is 0.13. This means that there is no enrichment of any transcription factors among differentially active sequences after appropriate multiple testing correction. I did not check this for the other analyses, but multiple testing correction should be performed over the entire analyzed dataset and not per cell type throughout the study, for instance for enrichment of GO, HPO etc.

Following the reviewer’s comment, we applied the FDR adjustment jointly on all three cell types in every analysis of this kind in the paper (TF binding motifs, predicted TF binding vs expression, Gene ORGANizer, HPO and Gene Ontology). Additionally, following comment #14, we have now removed from the analyses TFs that are not expressed in these cell types (see response to #14). Consequently, SP3 is no longer significantly enriched and was therefore removed from the “Molecular mechanisms underlying differential activity” section and from Supplementary Figure 4. ZNF281 remains significantly enriched (4.6-fold in NPCs, FDR = 0.04). Similarly, we have updated the Gene Ontology, Gene ORGANizer and HPO results (Figure 4a,b, and Supplementary Table 4). The top results we focused on in the Gene ORGANizer analysis (i.e., vocal cords, larynx, pharynx, and cerebellum) remain significant (FDR < 0.05), while the urethra is no longer significant and was therefore removed from the “Phenotypic consequences of differential expression” section and from figure 4a. For the HPO analysis, on top of applying FDR jointly on the three cell types, we have lowered the cutoff for the minimum number of genes per term from 5 to 3, because HPO terms tend to be linked to considerably fewer genes than GO and Gene ORGANizer and terms (Gene ORGANizer aggregates HPO terms into organs). Figure 4a,b, Supplementary File 4, and the “Phenotypic consequences of differential expression” section have all been updated with the new FDRs.

11. Line 551-557, This seems like a gross overinterpretation to me. If a binding site is detected or not given one base pair, is most closely related to the significance cutoff for detection of a binding site in the software being used. This could also be interpreted such that if it depends on one single base pair if there is binding or not, there might not be much binding at all. That the cell regulatory machinery uses the same cutoff for binding as the method to detect binding in this study is extremely unlikely. Further, binding is not only determined by the sequence itself but also by sequence activity and accessibility. In other words, the evolution of a new putative binding site in inaccessible heterochromatin is likely inconsequential. It would be more likely for a newly evolved, putative binding site to be active adjacent to an already existing, active enhancer, where the sequence would already be accessible. But lentiMPRA is not appropriate for detecting such events and speculations about newly evolved regulatory elements seem unwarranted.

We agree with the reviewer that this is a more complicated point than our description suggests. Therefore, we have removed this short paragraph from the discussion.

12. Line 621-622, How many of the variants included in the study have information in only one of the archaic human genomes?

For the design of this library, we took only sequences with information from all three high-coverage archaic genomes available when the library was designed (Altai and Vindija Neanderthals genomes, and the Denisovan genome). We later also removed sequences where a fourth archaic human genome (the Chagyrskaya Neanderthal) did not match the sequence in the other archaic genomes. To clarify this, we rephrased this section in the Methods to: “As a basis, we used the list of 321,820 modern human-derived single nucleotide changes reported to differ between modern humans and the Altai Neanderthal genome. We then filtered this list to include only positions where the Vindija Neanderthal and Denisovan sequences both match the Altai Neanderthal variant… and we subsequently also filtered out loci at which the Chagyrskaya Neanderthal genome did not match the ancestral sequence, bringing the final list of analyzed loci to 14,042”.

13. Lines 636-639, The way how the design of the oligos is described is very confusing and maybe actually misleading. For instance here: "For each variant we designed two fragments, one with the ancestral sequence and one with the derived sequence." I don't think that is correct. If I understand correctly, it should say "For each variant we designed two fragments, one with the ancestral variant state and one with the derived variant state, each in the background of the modern human [Is this correct?] sequence." because sequence versions do not differ in their entire sequence but only in the single variant whereas the sequences are otherwise identical.

Following this and comment #1 made by the reviewer, we realized this was unclear and rephrased this section. Please see our response to comment #1.

14. Line 905, Where TFs filtered for expression in the analyzed cell type? If not, that should be done!

We thank the reviewer for pointing this out. Following the reviewer’s comment, we removed from the analyses TFs that are not expressed in the cell types examined (FPKM <= 1). For ESC expression, we used ENCODE RNA-seq data for H1-hESC, downloaded from EBI Array Express under accession number: E-GEOD-26284. For osteoblast expression, we used the Moriarity et al., Nat Gen., 2015 RNA-seq data, downloaded from GEO under accession number: GSE57925. For NPC expression, we used the Lu et al. Mol. Cell, 2020 RNA-seq data, downloaded from GEO under accession number: GSE115407. This is now described in the Methods (“Differential transcription factor binding sites” chapter). The updated results appear in Supplementary Figure 5d and Supplementary File 3. Importantly, the TF we focused on in this section (ZNF281) is expressed in the relevant cell type (NPC).

15. Line 906-908, "For cases in which multiple unique motifs corresponded to the same TF, we used the motif with the largest score difference between alleles." I don't see why that would make sense. If there are different score differences for different motifs on the same allele, wouldn't the expectation be that in each case the highest score binds? That could mean that a substitution would swap one motif for another, such that the binding differential of either motif is irrelevant to the question but what really matters instead is the binding differential between the respective maximum binding affinity of the two motifs.

Following the reviewer’s comment, we now take for each motif its maximum predicted binding score in each of the two alleles and then take the score difference between these maximums. We updated the Methods (“Differential transcription factor binding sites” section) and the binding-expression correlation analysis in Supplementary Figure 5d accordingly (and there was no effect on the TF enrichment results).

Reviewer #2:Weiss et al. screened 14,042 variants that differ between modern and archaic humans genome-wide for their gene-regulatory impact. Using massively parallel reporter assay (lentiMPRA) in three cell types, they find evidence that ~13% of genomic sequences harboring variants can drive gene expression (=active sequences), and that about 13% * 23% ≈ 3% of variants lead to differential regulatory activity (=differentially active sequences). The authors study, characterize and compare active and differentially active sequences with regard to genome annotations; they find, for example, that differentially active sequences are enriched in annotated enhancer loci. With regards to a possible mechanism for differential regulatory activity observed, Weiss et al. study differences in transcription factor (TF) binding site motifs between corresponding archaic vs. modern sequences, and they are able to report substantial motif changes for ZNF81 and SP3. Linking first variants to genes and then second genes to organs and phenotypes allows the authors to relate putative variants' effects to brain and vocal tract; then a variant with reduced regulatory activity in the modern human version (across all there cell types studied) is discussed in more detail, with the conclusion that reduced SATB2 expression could give rise to brain and craniofacial phenotypes different between modern and archaic humans.Effects of (modern) human-specific variants are of great interest, so this study using powerful MPRA technology for a comprehensive assessment of all variants is exciting and a step forward. The authors are able to suggest specific variants that may underlie modern human traits, and overall the manuscript is clear. The linking of variants to organs and phenotypes is interesting, and it provides several potential entry points for follow-up studies. Importantly, the data provided constitute a significant resource for the community and will have an impact on further studies in this area.

We share the reviewer’s point of view that the catalog we provide and the trends we identify could serve as entry points and an important resource for the community for follow-up studies.

While it would be great to have a more definite linking of variants to genes, in general and specifically in the case of SABT2, where there is no direct experimental evidence that chr2:199,469,203 affects that gene's expression, that would require additional experiments. An easier improvement would be for the authors to publish the computer code used to generate results; currently this is not the case, nor do the authors state that it is made available upon (reasonable) request. This makes it very hard to reproduce results reported, study details of the analyses performed, or explore effects of parameter choices, thereby diminishing the value/impact of the study. The manuscript would benefit from addressing the following specific points.

We agree with the reviewer that further experimental evidence linking the chr2:199,469,203 variant to *SATB2* would be useful. We are currently working on using CRISPR/Cas9 to edit this variant and investigate its function and effect on *SATB2* expression. As the reviewer mentions, this is a relatively difficult task and we are unlikely to include it in the current study, especially considering the covid-19-related restrictions on wet lab work. It is also important to emphasize that several independent pieces of evidence suggest that this locus has a regulatory function that potentially affects *SATB2*: (i) This position is found within the putative promoter of *SATB2*, 4 kb upstream of the first TSS and 2 kb downstream of the second TSS, (ii) It is a relatively conserved position in vertebrates, found within a CpG island and a DNase I-hypersensitive site that shows active histone modification marks in all three cell types, and (iii) It is also found within a lncRNA that has been shown to regulate *SATB2* expression (Liu et al., 2017). To further clarify to readers that the link between this variant and the expression of *SATB2* remains to be determined, we toned down this section, which now reads: “Together, these data support a model whereby the C->T substitution in the putative promoter of *SATB2*, which emerged and reached fixation in modern humans, possibly reduced the expression of *SATB2”.*

Additionally, as the reviewer suggested, we have made the code fully available on Github (https://github.com/weiss19/AH-v-MH), and have added the following statement to the Methods: “Code is available for download on Github: https://github.com/weiss19/AH-v-MH”.

1. P8 / Supplementary Figure 1: Panel B: By eye the data looks quite a bit more noisy for ESC than for the other two cell types. Correlation is less for ESC, and seems driven by only few sequences. Additionally reporting Spearman correlation would be good, and I this should be discussed in the manuscript.

We agree with the reviewer that the ESC data appears noisier than the other two cell types. As per the reviewer’s suggestion, we calculated Spearman’s correlation for each comparison in Supplementary Figure 1b, and saw that while the correlation for each cell type decreased slightly, for ESCs the correlation decreased more substantially (from ~0.76 to ~0.48). Importantly, this correlation was still very significant (*P =* 1.8x10^-143^ to 2.4x10^-139^). We added Spearman’s r to the figure. We believe technical reasons could partially explain this with ESCs being very sensitive to infection (see methods where we note the MOI used for ESCs was 10 compared to 50 for the other cell types and that “We used a lower MOI for ESC because the cells are very sensitive to infection and a MOI higher than 10 would result in cell death”). Due to the difficulty in infection, the quality of the data was slightly lower for ESCs than for the other cell types. To address this, we added the following to the Results section: “We saw a strong correlation of RNA/DNA ratios between replicates for all cell types (Pearson’s r = 0.76 – 0.96, *P* < 10^-100^ , Supplementary Figure 2b), with the lower correlation scores being in ESC, likely due to our use of lower multiplicity of infection (MOI) in these cells due to their increased sensitivity to lentivirus infection”. We also note in the methods that we required additional sequencing for ESCs “due to lower lentivirus infection efficiency in this cell type.”

Panel D: It would be more informative using a log-scale for the RNA/DNA ratio. Also, it would be informative to see the tested sequences together with the scrambled and the positive.

We agree with the reviewer. Following the reviewer’s comment, we changed the scale to a log2 scale and added boxes for the tested sequences (split into active and inactive to show the distribution of each).

2. P9 L 169: It might be interesting for the authors to provide an overview of where/of what type the sequences are. For example., are they mostly intergenic, proximal, or inside genes? If so, is there perhaps an enrichment for certain types of genes, etc.? Even in case not novel, where archaic vs. modern human differences are would be good to know in light of contrasting active vs. inactive sequences later in the manuscript.

We agree with the reviewer that further information about the distribution of these variants was missing. Following the reviewer’s comment, we added to the paragraph introducing the 14,042 variants a description of their distribution, namely that “the vast majority of the variants are intergenic”, and also refer the readers to Supplementary Figure 1a, which describes the genomic annotation of these variants.

We have also added analyses of the enrichment of Gene Ontology, HPO and Gene ORGANizer terms within genes associated with these variants (Supplementary File 5) and added a description of these results to the “Phenotype enrichment analyses” section in the Methods. The top Gene Ontology terms associated with these genes are GO:0042472 *inner ear morphogenesis* (FDR = 2x10^-3^), GO:0007157 *heterophilic cell-cell adhesion via plasma membrane cell adhesion molecules* (FDR = 2x10^-3^) and GO:0048706 *embryonic skeletal system development* (FDR = 3x10^-3^). The top HPO terms associated with these genes are *Broad nasal tip* (FDR = 4x10^-6^), *Long philtrum* (FDR = 2x10^-4^), and *X-linked inheritance* (FDR = 3x10^-4^). The Gene ORGANizer analysis did not provide any significant results. Importantly, unlike the analyses of differentially active sequences, which were compared against a background of non-differentially active sequences to control for potential confounders, the full set of genes in this new analysis is not compared against a background. Therefore, these results may be affected by different confounders such as GC content, the ability to call SNPs, and DNA degradation patterns in ancient samples, and it is still to be determined to what extent these results reflect true evolutionary trends.

3. P9 L 172 / Methods P38-40 L761-802: While the authors describe their MPRA analysis, safe for actually attempting to re-do the analysis, it remains unclear whether sufficient details are provided to reproduce results; publishing code used for the analysis would be far better. Further on, processed data provided on GEO has replicates already aggregated, so that it cannot be used to reproduce, e.g., the MPRAnalyze part of the analysis and primary data would have to be re-processed first.

We agree with the reviewer that it is important for code to be made publicly available for readers of the manuscript. As such, we have created a Github repository to house the code for the project: https://github.com/weiss19/AH-v-MH.

We have also added the following line to the Methods: “Code is available for download on Github: https://github.com/weiss19/AH-v-MH.”.

Also, it is unclear why the authors use a combination of MPRAnalyze and a heuristic involving aggregation across replicates to assign oligos/sequences as "active". It'd be good to explain why that is done, and what is the effect on the set of active oligos.

Following the reviewer’s comment, we realized that the reasoning behind using two approaches to determine if a sequence is active was not clear enough. We applied two separate tests in order to add another layer of stringency to this analysis. Further stringency was required because MPRAnalyze did not seem to completely account for the lower lentiviral infection efficiency in the ESC library, which reduced library complexity, resulting in a possible overestimation of active sequences in these cells (i.e., twice or more active sequences identified in ESCs compared to the other cell types: 2,097 in ESCs, 1,059 in osteoblasts and 664 in NPCs). We therefore decided to apply another test of activity, as used in previous MPRA analyses, where the 95th percentile of the scrambled sequences expression is used as a cutoff for activity (see for example Kwasnieski et al., Genome Research, 2014). Indeed, almost half of the ESC active sequences did not pass the second test, while sequences in the other cells were not as affected (1,183 sequences passed both tests in ESCs, 814 in osteoblasts and 602 in NPCs). We now explain this in more detail in the “Measurement of expression and differential expression” section in the methods: ” Next, we applied a second test for activity, to account for potential overestimation of active sequences in ESCs due to the lower lentiviral infection efficiency in these cells.”

4. P10 L 190-192: Figure 2: Panels B-D: For each bin of RNA/DNA ratio enrichment appears to be plotted, based on a hypergeometic/Fisher test, probably as the logarithm of the odds ratio. It would be informative to plot a measure of variability/confidence for each bin as well. Standard implementation of the Fisher test in R provides a confidence interval, for example.

We agree with the reviewer that confidence intervals would be informative for these results. We have tried to add them to the figure itself but it made the figure very busy and hard to read (as each panel included over 40 confidence intervals, which overlap with one another at each x-axis cutoff). We therefore added this information to Supplementary File 2 and refer to the table in the legend of the figure.

5. P10 L 198-200, Figure 2: Similar to the point above, it would be good to see "error bars" on panel e.

Similar to the previous point, we added confidence intervals to the table, as the figure became too busy with overlapping whiskers.

6. P12 L 235: It is unclear what the null model for the Super Exact test is. Is overlap of differentially active sequences between cell types higher than expected, given what we already know about the overlap considering all active sequences? Or do cell types overlap for differentially active sequences more than expected given sequences were selected purely at random, but not as much as they do considering all active sequences?

The Super Exact test for the overlap of differentially active sequences was done in comparison to the active sequences as background. Following the reviewer’s comment, we realized that this was unclear from the text and we added this information to the legend of Figure 3d and the main text: “8-fold higher than expected compared to active sequences (P = 5x10^-7^, Super Exact test, Figure 3d).”

7. P15 L 272-274: Supplementary Figure 3: Variability between lentiMPRA replicates is not displayed in panel B; it would be good to use a display that shows variability in replicates of both arrays, lentiMPRA and luciferase assay.

We agree. Following the reviewer’s comment, we added per-replicate data to the figure (now Supplementary Figure 3c).

8. P16 L 300-305: For the methylation analysis, it there a cell type signal in the sense that up-regulation correlates with hypomethylation and down regulation with hypermethylation in osteoblasts, but not in the other two cell types?

The trend is observable also in the NPCs (e.g., the top 10 upregulating sequences are hypomethylated by 7% on average in modern compared to archaic humans, the top 10 downregulating sequences are hypermethylated by 13% in modern compared to archaic humans). This is in line with previous observations that differentially methylated regions tend to be shared across tissues (e.g., Hernando-Herraez et al., Dynamics of DNA methylation in recent human and great ape evolution, PLoS Genet., 2013). In the ESC data we do not see this correlation, perhaps due to its noisier nature in this experiment. We have added this to the “DNA methylation in active and differentially active sequences” section in the Methods.

9. P17 L 322-325: Supplementary Figure 4: What exactly is plotted in panels B-D: does each point correspond to a sequence-TF combination, or was some kind of averaging performed (over sequences or TFs)?

Each point in these plots corresponds to a single sequence, i.e., its expression fold-change (x-axis) and predicted TF binding difference (y-axis). To clarify this, we added “for each sequence” to the figure legend.

10. P44 L 900-903: FIMO analysis: Unclear why the lowest-matching score would be used. Why not use the highest log odds on that allele? Also, what background model was used for FIMO?

A background model for FIMO was generated using fasta-get-markov using the trimmed (or untrimmed, if >1 variant) sequences. As for the score difference calculation, FIMO uses a p-value cutoff of 10^-4^ for reporting predicted binding (https://meme-suite.org/meme/doc/fimo.html). Therefore, some sequence pairs have a reported score for only one of the alleles. To assign these sequence pairs with a score difference, we used a conservative approach where we assigned the unscored allele with the lowest score reported for that motif, representing a score that is closest to a *p*-value of 10^-4^. Because the unreported score could be anywhere below this lowest reported score, but could not have been above it, this results in a conservative underestimation of the score difference. Then, as in other sequence pairs and as the reviewer suggested, we took the highest score difference per TF. To clarify this, we rephrased this section in the Methods and also added the information about the background model (see the “Differential transcription factor binding sites” section in the Methods).

11. P18, L 342: How sensitive are the results of the enrichment analysis (ZNF281 and SP3) with regards to some of the parameters, like requiring that 10 or more motif changes are required for TFs to be considered? How many TFs did meet that requirement?

Following the changes we have made to this analysis based on comment #10 by Reviewer 1 (carrying out multiple testing adjustment across all three cell lines), SP3 is no longer significantly enriched in this analysis. We therefore examined the enrichment of ZNF281 across various cutoffs of the number of predicted bound motifs, ranging from 5 to a maximum of 14 (the number of motifs predicted to be differentially bound by ZNF281) in steps of 1. We found that with the exception of the cutoffs of 5 and 6 (where ZNF281 is only slightly above the significance threshold: FDR = 0.058 and FDR = 0.053, respectively), ZNF281 is the only significant TF across all of these cutoffs (FDR ≤ 0.05). Overall, 27 TFs pass the minimum cutoff of 10 predicted binding motifs that we used.

We also tested the other parameter used in this analysis: the FPKM level of each TF. We repeated the analysis for FPKM minimum cutoffs ranging from 0.5 to 3 in steps of 0.5, and found that ZNF281 is significantly enriched across all of these cutoffs, and is the only significantly enriched TF (FDR ≤ 0.05). We now describe this in the “Differential transcription factor binding sites” section in the Methods.

12. P20 L 371-376: In the supplementary table, it would be good to provide information about how a gene was linked to a variant (i.e., which of the four lines of evidence used support the link). Also, for the Hi-C data, the authors describe two lists in the methods a cell type-specific list and a general list. However, it appears unclear which one was used for linking to genes.

Following the reviewer’s suggestion, we added to the “Linked genes” sheet in Supplementary File 1 the information on the source through which each sequence was linked to its genes (see the “sources for sequence-gene links” columns). With regard to the type of Hi-C data used, for the cell-type-specific (stringent) list, we used the cell-type-specific Hi-C links. For the generic (non-stringent) list, we used the general list. In Supplementary File 1 we report both lists, but for the enrichment analyses we used the generic list to have more statistical power. To address this, we rephrased these sections in the Methods, which now read: “For the cell type-specific (stringent) list of locus-gene links, we included only those interactions observed in cell types corresponding to the cell lines used in our lentiMPRA: ESCs, NPCs and mesenchymal stem cells as an approximation for osteoblasts (given that osteoblast Hi-C data is not publicly available to the best of our knowledge, and that osteoblasts differentiate from MSCs). For the generic (non-stringent) list, we used interactions across any of the 27 tissue and cell types analyzed by Jung et al.” and “We conducted these analyses.… on the non-stringent locus-gene associations”.

13. P20 L 387-391 and P21 L 394-396: Which genes are driving theses terms? Do they largely overlap between different terms, or are they different genes in each term. It would be informative to report genes alongside terms in the supplemental table. Also for the HPO analysis in the following paragraph.

Following the reviewer’s comment, we added to Supplementary File 4 the lists of genes associated with each term in each of the enrichment analyses (HPO, Gene ORGANizer and Gene Ontology). With regard to the overlap of genes between different terms: in terms that are tightly associated with one another (e.g., HP:0000089 – *Renal hypoplasia* and HP:0000076 – Vesicoureteral reflux) we observe the expected overlap (*UFD1, COMT, RAD21* for *Renal hypoplasia* and *UFD1, COMT, RAD21,*and *CDC45* for *Vesicoureteral reflux*). For terms that are less tightly associated with one another (e.g., HP:0000028 – *Cryptorchidism* and HP:0000488 – *Retinopathy*), we see few to no overlapping genes (*DGCR6, CDC45, UFD1, DGCR8, TXNRD2, COMT, RAD21* for *Cryptorchidism* and *PTPN22, ERF2IP, TMEM231,*and *NDUFAF3* for *Retinopathy*).

14. P23 L 432: PyloP conservation scores are usually -log10(p-value), so while conservation is supported, the p-value corresponding to a score of 0.996 would be about 0.1, which is not that highly conserved.

Indeed, using “highly conserved” was inaccurate and we therefore changed it to “relatively conserved”.

15. P25 L 484-486. The authors write they have investigated how single nucleotide variants "have affected expression". While technically true in the context of the lentiMPRA, the authors measure changes in the ability of these sequences to drive expression, but do not show that they actually affect "real live" gene expression outside of an artificial construct. Perhaps change this sentence.

We agree that we cannot claim that these variants have necessarily affected expression in their native genomic locations. Therefore, we have rephrased this sentence and this line now reads “By comparing modern to archaic sequences, we investigated the regulatory potential of each of the 14,042 single-nucleotide variants that emerged and reached fixation or near fixation in modern humans”.

Reviewer #3:The majority of sequence variants that separate modern and archaic human genomes are found in non-coding regions, often overlapping gene regulatory elements. It is difficult to study the functional significance of these variants on molecular phenotypes such as transcription and chromatin regulation due to the lack of intact biological material in archaic specimens. The current work uses an MPRA framework to compare the gene regulatory activities of derived modern human versus ancestral variants in three modern human cell types – ESCs, NPCs and osteoblasts. As expected, the authors found that active sequences were associated with enhancer annotations, and differentially active sequences were associated with divergent TF binding motifs. The presented MPRA data is well controlled. However, the differences detected by the various comparisons made were often subtle, or small in magnitude. This requires robust statistical analysis to understand the significance of those differences. While the authors were moderate in most of their claims, this reviewer thinks some of those claims require further analytical support with some adjustments in the text. With these additional non-experimental, but critical analytical revisions and some clarifications, this would be of potentially high interest to a broad readership as a contribution to eLife.

We thank the reviewer for their constructive feedback and agree that this work has the potential to be of high interest to the readers of *eLife*. We have added several analyses and clarifications following the reviewer’s comments.

1. Selection of cell types – this is crucial as TFs are the major drivers of MPRA activity. The claim that ESCs were used "due to their globally active transcription, providing a general view of gene expression" is based on a single study from mouse embryonic stem cells (Efroni et al., 2008). Mouse ESCs are known to have very different biological properties from human ESCs, with mouse ESCs representing a less differentiated state than human ESCs (Hanna et al., PNAS 2010). Therefore, this is not a sufficient justification for using ESCs as a comparative cell type. In particular, enhancers are typically cell type or lineage specific in their activity, and they are driven by TFs that are expressed in a lineage dependent manner. hESCs express basal and pluripotency factors, limiting their capacity to drive transcription of developmentally specialized enhancers. The use of NPCs and osteoblasts are reasonable considering the central interest of the study in understanding the role of derived alleles in shaping modern human phenotypic differences, but beyond that point it is not clear why the cell types were chosen (see point 2).

We are glad the reviewer found our use of NPCs and osteoblasts to be reasonable and within the scope of our project. Additional reasons for choosing these cell types were the abundance of previously published datasets for these cell types (e.g., chromHMM) that can be used to uncover regulatory patterns, their stability in culture and infection, and the ability to compare osteoblast data with ancient DNA methylation maps derived from bone samples.

With regard to ESCs, hESCs indeed show many characteristics that distinguish them from mESCs, and the field seems to be split on the exact characteristics of the transcriptome in hESCs. In addition to Efroni et al., 2008, which we cite in our manuscript, there are additional sources that suggest global transcription is a hallmark of embryonic stem cells (e.g., Gulati, et al. 2020 doi: 10.1126/science.aax0249, Egli). There is, however, also evidence that ESCs have their own unique transcriptome that changes through differentiation (Inoue, et al. 2019 doi: 10.1016/j.stem.2019.09.010), and as the reviewer mentioned, the extent to which mESCs and hESCs share transcriptional characteristics is still under debate. Since there is some disagreement as to whether ESCs show global transcription or not, we removed this reasoning from the manuscript, and instead present the other reasons why these cells were picked, namely because: (i) stem cells provide unique insight to an early developmental stage in human cells, which is commonly used to interrogate human evolution and early development (Blake, et al. 2018 doi: 10.1186/s13059-018-1490-5; Loh, et al. 2014 doi: 10.1016/j.stem.2013.12.007; Mora-Bermúdez, et al. 2016 doi: 10.7554/*eLife*.18683; Nuttle, et al. 2016. doi: 10.1038/nature19075), (ii) the NPCs we used were derived from this H1-ESC line, and (iii) ESCs are extensively characterized, providing a wealth of data available for these cells (e.g., chromHMM and mapping of bivalency). The text now reads “The brain and skeleton have been the focus of evolutionary studies due to their extensive phenotypic divergence among human lineages. Therefore, we chose human cells related to each of these central systems: NPCs and primary fetal osteoblasts. In addition, we used ESCs (line H1, from which the NPCs were derived) to gain insight into early stages of development. Finally, the abundance of previously published regulatory maps for these three cell types also enables the investigation of the dynamics of evolutionary divergence at different regulatory levels. While these cell types represent diverse systems, further studies are needed in order to characterize the activity of these sequences in other cell types.”.

2. Of the 14000 regions selected for the MPRA – what was the expectation given existing functional genomic data from modern humans and how did that inform the selection of cell types? This question is related to the above point. In supplementary figure 2 the authors provide the annotation classifications of the 14,000 fixed derived variants based on ChromHMM. This data indicates that ~25% of these variants overlap enhancers or TSS regions – but the relative distributions of the annotations differ depending on the ChromHMM context (ESC, NPC or Osteoblasts). It seems that using this methodology, one can predict other cell-types for which these variants may be active, thus providing an informed context to test the MPRA. It is important to know what percentage of the fixed variants are located in any type of regulatory region across a variety of cell-types. This information may also be important for understanding differential activity of modern vs archaic sequences, as the magnitude of differential activity may be cell type dependent.

We agree with the reviewer that chromatin marks in other cell types could help shed light on the activity of these sequences in additional cell types that were not included in this work, and that this information was missing from the manuscript. We therefore added to the manuscript chromHMM annotations and analyses in ten additional tissues/cell types. On top of ESCs, osteoblasts, and NPCs, these samples include mesenchymal stem cells, monocytes, skin fibroblasts, brain hippocampus, skeletal muscle, heart left ventricle, sigmoid colon, ovary, fetal lung, and liver. We picked these tissues to represent a wide range of regulatory programs, and to take samples where the Roadmap reported quality was defined as the highest (i.e., 1). The chromHMM annotations for these 13 tissues now appear Supplementary Table 1. We used these annotations to explore which of the 14,042 sequence pairs in this study show active chromHMM annotations (i.e., enhancer- and TSS-related annotations) and in what tissues. We found that 4,821 out of the 14,042 (34%) sequence pairs overlap active chromHMM annotations in at least one tissue, with 740 of the sequence pairs (5%) overlapping active chromHMM annotations in over half of the tissues. We now discuss this in the main text. We have also added a panel to Supplementary Figure 1 showing a histogram of these results, as well as the overlap within the three cell types used in this study. We hope this new data would be informative of potential activity of these sequences in tissues not included in this study.

Interpreting our results in light of previous MPRAs is challenging, not only because of key differences in statistical power and experimental design (e.g., sequence length), but also because of the way sequences were selected for each MPRA. Previous reporter assays and MPRAs on human variation used pre-filtered sets of variants by selecting sequences with putative regulatory function (e.g., eQTLs, TF binding sites, ChIP-seq peaks, or TSSs) and/or regions showing particularly rapid evolution (e.g., human accelerated regions, see Discussion for references). In our experiment, on the other hand, a key consideration was not to pre-filter the pool of sequences based on their potential activity and/or cell type specificity, but rather generate an unbiased scan of fixed modern human-derived variants. This allowed us to measure regulatory activity in regions that would otherwise be excluded by filtering for a specific set of marks, and to get a relatively unbiased estimate of the proportion of fixed changes that have a potential regulatory effect. In line with the fact that our data was not pre-filtered for putative regulatory regions, the proportion of active sequences we observed tended to be slightly lower than in previous studies. However, the magnitude of differential expression we observed, as well as the fraction of differentially active sequences out of the active sequences was similar to previous studies. For example, Tewhey et al., 2016 investigated 32,373 variants previously identified as *cis*-eQTLs in humans, and found that 12% of these sequences showed activity in their assay, and that 25% of them showed differential activity, with only 46 sequences showing a 2-fold change or higher. Uebbing et al. 2021 investigated 32,776 substitutions in human accelerated regions and human gain enhancers and found that 11.7% of HARs and 33.9% of HGEs were active in neurodevelopment, of which 27.5% and 34.6%, respectively, were differentially active between human and chimpanzee, with a mean fold-change of 1.58x. To address this, the text now reads: “In line with the fact that our data was not pre-filtered for putative regulatory regions, the proportion of active sequences we observed tends to be slightly lower than these previous studies. However, the magnitude of differential activity, as well as the fraction of differentially active sequences out of the active sequences was similar to previous studies^16,28–31,83–85^“. We trust that future unfiltered MPRA studies will help shed light on how these modern-archaic human differences compare to intra- and inter-species effects of single-nucleotide changes.

Related to this point, in the Discussion the authors point out that, based on previous studies, trans differences are likely minimal between modern and archaic humans, but what about trans differences in cell-type environment? Their assertion is perhaps an overgeneralization that diminishes the importance of trans effects. This is a point that should be addressed.

We agree with the reviewer that this sentence was an overgeneralization and that it diminished the importance of trans effects. Trans differences between cell types are indeed substantially more pronounced than trans differences between the same cell type of two human groups. Because we did not compare the activity of the modern human allele in one cell type to the archaic human allele in another cell type, these trans differences should not cause false positives in our results. However, trans differences between cell types very likely underlie the observation that differential activity in one cell type is not always shared with other cell types. We now realize that the statement was an overgeneralization and our intent was not to downplay the role of trans effects. We therefore rephrased it to: “Finally, differences in the trans environment of a cell could have an effect on the ability of a sequence to exert its cis*-*regulatory effect, resulting in cell-type-specific cis-regulatory effects, as we observed in our data. The trans environment of the same cell type might also differ between two organisms. However, the majority of the cis-regulatory changes we observed would be expected to be present in archaic human cells as well, considering that such conservation has been observed between substantially more divergent organisms (e.g., human-chimpanzee Ryu et al., 2018 and human-mouse Köhler et al., 2014). In other words, while trans-regulatory changes play a key role in species divergence, the trans environments of the same cell type in two closely related organisms tend to affect cis-regulation similarly.”

3. The authors report a modest median fold-change of 1.2x for differentially active sequences. They use a combination of criteria to associate these sequences with a putative target gene. An important next step to support the significance of these findings would be to understand whether these modest differences are linked to differential expression of target genes. Obviously, such a comparison cannot be made for modern and archaic human transcriptomes; however, one potential avenue for testing this idea would be to use public data from comparative human and chimp RNA-seq studies to look at whether the target genes are differentially expressed between modern human and chimp, provided that the ancestral allele for the differentially active sequence is found in chimp.

Following the reviewer’s comment, we investigated the expression of genes associated with differentially active sequences using human and chimp RNA-seq data. As the expression changes we report are driven by *cis*-regulatory changes, we used our recently generated RNA-seq data from human-chimp hybrid cells (Gokhman et al., 2021, Nature Genetics, in press, see read count and raw sequencing data in GEO accession numbers: GSE146481 and GSE144825). In these hybrid cells, the human and chimpanzee chromosomes are found within the same nuclear environment and are exposed to the same *trans* factors (e.g., transcription factors). Therefore, any differential expression observed between the human and chimpanzee alleles within these hybrid cells is attributed to *cis*-regulatory changes. The cells we generated are hybrid human-chimp induced pluripotent stem cells (iPSCs), and we therefore investigated whether genes associated with upregulating sequences in our ESC MPRA data tend to be upregulated in the hybrid iPSCs, and vice versa. It is important to note that differential expression between humans and chimpanzees reflects ~12 million years of evolution (i.e., changes that emerged along the human as well as along the chimpanzee lineages since their split from their common ancestor ~6 million years ago). However, our MPRA data was done on sequences that changed along the modern human lineage (~550-765 thousand years). Therefore, the human-chimp differences span an evolutionary time that is ~20-fold longer than the modern human lineage, and the effect of modern-derived variants on gene expression between humans and chimpanzees is expected to be largely diluted by the many other changes that accumulated along the rest of this time. Indeed, we observe a very slight, but significant correlation between differential expression observed in the MPRA data and differential expression observed in the human-chimp hybrid data (*P* = 0.017, Pearson’s r = 0.1).

These results now appear in a new section in the Methods titled: “Human-chimpanzee cis-regulatory expression changes”.

4. The section describing the relationship between DNA methylation and differentially active sequences seems to be one of the least well supported pieces of this study. In the absence of gene expression data, a key piece of support for the functional significance of differentially active sequences is to link those differences to epigenetic differences. The availability of DNA methylation maps from Neanderthal and Denisovan makes this possible. However, the way this data is presented (or not presented) is really difficult to interpret (i.e. reporting that upregulating sequences tend to be hypomethylated with a -1% difference compared to what?). A figure would have been very useful here.

Following the reviewer’s comment, we have made several changes to this paragraph. First, we reworded this section to clarify the comparison that was done. In short, the comparison was done between sequence pairs, i.e., for each pair of modern and archaic human sequences we tested if the human lineage showing reduced activity in the lentiMPRA is also the human lineage showing hypermethylation of that sequence in its native genomic context. For example, if a sequence is downregulating in modern compared to archaic humans, we tested if it is also hypermethylated in modern compared to archaic humans. The results we found show a weak, but significant, association of increased methylation and reduced activity (i.e., downregulating sequences show on average 2% more methylation in modern compared to archaic genomes, and upregulating sequences show 1% less methylation). The trend is slightly stronger when looking at the top 10 most divergent sequences (-7.4% and +8.3%, respectively). Following the reviewer’s suggestion, we added a violin plot showing the methylation levels for these sequences (Supplementary Figure 4a). Additionally, because methylation is known to be most tightly linked with activity in promoter regions, we added a section examining the relationship between differential methylation and differential activity of sequences within promoter regions. Here too, we found that sequences upregulating activity in modern humans are hypomethylated by 5% on average in the modern compared to the archaic genomes, while downregulating sequences are hypermethylated by 8% on average (*P* = 0.034, paired *t*-test). These results now appear in Supplementary Figure 4b. We have also toned down this paragraph and removed the sentence about the overlap with differentially methylated regions, as it was both confusing and was only based on four observations. Finally, to further investigate the relationship between DNA methylation and activity, we added a paragraph to the “Characterization of active regulatory sequences” section, where we explore the link between the two. We found that active sequences are significantly hypomethylated compared to inactive sequences (*P* = 5.5x10^-13^, *t*-test) and that their activity level (RNA/DNA ratio) is negatively correlated with methylation levels (*P* = 6.0x10^-9^, Pearson’s r = -0.24). We now show these new results in Figure 2f.

Also, it's important to note that many enhancers have low CpG density, while promoters and TSSs with high CpG density are typically hypomethylated regardless of gene activity (Lister et al. 2009, Stadler et al. 2011, Schlesinger et al. 2013). Thus, it is necessary to parse these regions when referring to methylation differences.

We thank the reviewer for this suggestion. Indeed, CpG composition varies across different functional parts of the genome and this could potentially be correlated with methylation levels. Following the reviewer’s comment, we added several tests to the differential methylation analysis section.

First, we ranked differentially active promoter sequences based on their CpG density and tested whether the hypomethylation we observe in upregulating sequences and the hypermethylation we observe in downregulating sequences become more pronounced in promoters with lower CpG density. Indeed, as the reviewer mentioned, the link between methylation and activity is stronger in CpG-poor promoter sequences: within upregulating promoter sequences, when taking those ranking at the bottom half based on their CpG density, they are hypomethylated by 15% on average in modern compared to archaic genomes (compared to 5% in all promoters, see above), while downregulating sequences show 15% hypermethylation (compared to 8% in all promoters, *P* = 0.006, paired *t*-test). We added these results to the methylation section and to Supplementary Figure 4c.

Second, to further test the potential effect of CpG density on our results, we compared CpG density in differentially active compared to non-differentially active sequences, and in upregulating compared to downregulating sequences. We found no significant difference in CpG density between these groups (*p*-values > 0.05, *t*-test). These results now appear in the “DNA methylation in active and differentially active sequences” section in the Methods.

Third, we tested chromHMM putative enhancer sequences and found a similar, but slightly weaker, trend compared to promoters, with 3% hypermethylation of downregulating sequences and 5% hypomethylation of upregulating sequences. Perhaps in accordance with the weaker link between enhancer methylation and activity, this trend is not significant despite having similar statistical power to the promoter analysis (*P* = 0.12, paired *t*-test). These results now appear in the “DNA methylation in active and differentially active sequences” section in the Methods.

Finally, it is important to note that the tests for hypo/hypermethylation were done in a pairwise manner by comparing the modern sequence to the archaic sequence, which are generally matched in their CpG density. Therefore, the decreased methylation we observe in upregulating sequences, and the increased methylation we observe in downregulating sequences are unlikely driven by differences in CpG density.